



# The SPARC water vapour assessment II:

# Profile-to-profile and climatological comparisons of stratospheric δD(H₂O) observations from satellite

Charlotta Högberg[1], Stefan Lossow[2], Ralf Bauer[3], Kaley A. Walker[3], Patrick Eriksson[4], Donal P. Murtagh[4], Gabriele P. Stiller[2], Jörg Steinwagner[5] and Qiong Zhang[1]

[1]Stockholm University, Department of Physical Geography and the Bolin Centre for Climate Research, Svante Arrhenius väg 8, 10691 Stockholm, Sweden.

[2]Karlsruhe Institute of Technology, Institute for Meteorology and Climate Research, Hermann-von-Helmholtz-Platz 1, 76344 Leopoldshafen, Germany.

[3]University of Toronto, Department of Physics, 60 St. George Street, Toronto, Ontario M5S 1A7, Canada.

[4]Chalmers University of Technology, Department of Earth and Space Sciences, Hörsalsvägen 11, 41296 Göteborg, Sweden.

[5]Max-Planck-Institute for extraterrestrial Physics, Gießenbachstraße 1, 85748 Garching, Germany.

Correspondence to: Charlotta Högberg (charlotta.hogberg@natgeo.su.se)





# Abstract.

Within the framework of the second SPARC (Stratosphere-troposphere Processes And their Role in Climate) water vapour assessment (WAVAS-II), we have evaluated five data sets of $\delta D(H_2O)$ obtained from observations of Odin/SMR (Sub-Millimetre Radiometer), Envisat/MIPAS (Environmental Satellite/Michelson Interferometer for Passive Atmospheric Sounding) and SCISAT/ACE-FTS (Science Satellite/Atmospheric Chemistry Experiment-Fourier Transform Spectrometer) using profile-to-profile and climatological comparisons. Our focus is on stratospheric altitudes, but results from the upper troposphere to the lower mesosphere are provided. There are clear quantitative differences in the measurements of the isotopic ratio, which primarily concerns the comparisons to the SMR data set. In the lower stratosphere, this data set shows a higher depletion than the MIPAS and ACE-FTS data sets. The differences maximise close to 50 hPa and exceed 200 per mille. With increasing altitude, the biases typically decrease. Above 4 hPa, the SMR data set shows a lower depletion than the MIPAS data sets, on occasion exceeding 100 per mille. Overall, the $\delta D$ biases of the SMR data set are driven by HDO biases in the lower stratosphere and by $H_2O$ biases in the upper stratosphere and lower mesosphere. In between, in the middle stratosphere, the biases in $\delta D$ are a combination of deviations in both HDO and $H_2O$. These biases are attributed to issues with the calibration, in particular in terms of the sideband filtering for $H_2O$, and uncertainties in spectroscopic parameters. The MIPAS and ACE-FTS data sets agree rather well between about 100 hPa and 10 hPa. The MIPAS data sets show less depletion below about 15 hPa (up to about 30 per mille), due to differences in both HDO and $H_2O$. Higher up the picture is reversed, and towards the upper stratosphere the biases typically increase. This is driven by increasing biases in $H_2O$ and on occasion the differences in $\delta D$ exceed 80 per mille. Below 100 hPa, the differences between the MIPAS and ACE-FTS data sets are even larger. In the climatological comparisons, the MIPAS data sets continue to show less depletion than the ACE-FTS data sets below 15 hPa during all seasons, with some variations in magnitude. The differences between the MIPAS and ACE-FTS data come from different aspects, such as differences in the temporal and spatial sampling (except for the profile-to-profile comparisons), cloud influence, vertical resolution, and the microwindows and spectroscopic database chosen. Differences between data sets from the same instrument are typically small in the stratosphere.



# 1 Introduction

Water vapour is one of the most important trace constituents in the Earth's atmosphere. In the upper troposphere and lower stratosphere, water vapour is the most important greenhouse gas. A large part of the predicted global warming is a result of different feedback processes induced by greenhouse gas emissions, where a significant contribution is related to an increased amount of water vapour in the troposphere due to increased average global temperatures. A warmer climate can hold more water vapour in the atmosphere and the main source for water vapour in the troposphere comes from evaporation, to a large part originating from the oceans. Changes in the troposphere will also influence higher altitudes, and it has been shown that the amount of stratospheric water vapour increases with increasing tropospheric temperature due to increased transport through the tropical tropopause layer (TTL; Gettelman et al., 2009). The strength of the stratospheric water vapour feedback, which implies that an increase of water vapour in the lower stratosphere in turn leads to an even warmer climate, is estimated to be 0.3 $Wm^{-2}$ for a 1 K temperature anomaly at 500 hPa (Dessler at al., 2013). As a main component of polar stratospheric clouds (PSCs), water vapour also plays a crucial role in the ozone chemistry in the middle atmosphere. The heterogeneous reactions that take place on the surface of the PSCs cause the ozone depletion observed during winter and spring in the polar lower stratosphere. The formation of PSCs also plays a role in dehydration in the polar regions during winter and spring, where large particles of the PSC type II containing ice crystals can sediment due to gravitational effects and permanently remove water vapour from the stratosphere (Kelly et al., 1989). Further water vapour is the primary source for the hydrogen radicals OH, H, and $HO_2$, which also contributes to the loss of ozone within auto-catalytic cycles. They dominate the ozone budget in the lower stratosphere and at altitudes above 50 km (Brasseur and Solomon, 2005). Beyond that, water vapour is a valuable tool to diagnose dynamical processes in the stratosphere and mesosphere (e.g. Mote et al., 1996; Seele and Hartogh, 1999; Lossow et al., 2009).

One of the main sources of stratospheric water vapour is the transport from the troposphere, which occurs mainly through the TTL. Slow ascent, accompanied by large horizontal motions, is thought to be the most important pathway (Fueglistaler et al., 2009). The amount of water vapour entering the stratosphere is controlled by the cold point temperature along the air parcel trajectories, which have a seasonal variation. Lower temperatures during winter time result in drier air entering the stratosphere. In the summer the situation is reversed, and





due to higher temperatures moister air enters the stratosphere. This signal is transported upwards by the upwelling branch of the Brewer–Dobson circulation and is preserved up to about 30 km before it dissipates. It is known as the tape recorder signal (Mote et al. 1996). Another pathway into the stratosphere is the convective lofting of ice (Moyer et al., 1996).

Typically 3.5 ppmv to 4.0 ppmv of water vapour are transported from the troposphere into the stratosphere (Kley et al., 2000). Within the stratosphere, the in situ oxidation of methane is the main source for water vapour. In general, one methane molecule produces two water vapour molecules (le Texier et al., 1988). With increasing altitude, the methane oxidation becomes more important and contributes the most to the water vapour budget in the upper

stratosphere. A minor source of water vapour is the oxidation of molecular hydrogen, which is of importance only in the upper stratosphere and lower mesosphere (Wrotny et al., 2010). The reaction with $O(^1D)$ to form hydrogen radicals is the major sink of water vapour in the stratosphere. With increasing altitude, the destruction by photolysis become more important. Overall, in the stratosphere the amount of water vapour increases with altitude. Around the

stratopause, a maximum is found. In the mesosphere the amount typically decreases, as there is no major source.

More than 99.7 % of water vapour exists in the form of $H_2^{16}O$ (hereafter named $H_2O$). There are several minor isotopologues, like $H_2^{18}O$ (0.20 %), $H_2^{17}O$ (0.037 %), and $HD^{16}O$ (0.03 %). Despite their low abundance, these minor isotopologues are important for atmospheric

science, as they eventually can provide additional information in the form of isotopic ratios relative to the main isotopologue, $H_2^{16}O$. In this regard, $HD^{16}O$ (hereafter named HDO) is most interesting, as the isotopic ratio typically exhibits pronounced variations. The isotopic ratio between HDO and $H_2O$ is typically given in the δD notation (Eq. 1), which describes the relative deviation of deuterium (D) to hydrogen (H) with respect to the reference ratio $R_{reference}$

= VSMOW = $155.76 \times 10^{-6}$ (Vienna Standard Mean Ocean Water, Hagemann et al., 1970).

$$\partial D(H_2O) = \left( \frac{R_{sample}}{R_{reference}} - 1 \right) \cdot 1000 \text{ ‰} \tag{1}$$

$$R_{sample} = \left( \frac{[D]}{[H]} \right)_{sample} \equiv \left( \frac{[HDO]}{2 \cdot [H_2O]} \right)_{sample} \tag{2}$$

Hereafter we refer to δD(H$_2$O) simply as δD. In Eq. (2), two approximations are made: (a) that the deuterium content in a sample is dominated by the contribution from $HD^{16}O$ and that

the contributions from the other deuterium bearing isotopologues are negligible, (b) that the



hydrogen content essentially comes from $H_2^{16}O$. In the upper troposphere and tropopause region, the isotopic ratio is primarily determined by condensation and evaporation processes, where the heavier isotopologue HDO is removed to a larger extent than $H_2O$ (vapour pressure isotope effect), leading to a depletion of heavier isotopologues with increasing altitude. In the

stratosphere, the oxidation of methane causes an increase in the isotopic ratio, since methane is not depleted in the heavier isotopologues to the same extent as water vapour during the transport from the troposphere to the stratosphere.

Given these influences, the isotopic ratio can be used to investigate the relative importance of different processes that contribute to the transport of water vapour from the troposphere to the

stratosphere (Moyer et al., 1996). If the dehydration due to the slow ascent of air through the TTL is considered on its own (which corresponds to a Rayleigh fractionation process), a δD value around −900 per mille would be expected near the tropopause. Observations, however, exhibit δD values between −700 per mille and −500 per mille (e.g. Johnson et al., 2001; Webster and Heymsfield, 2003). This indicates that non-Rayleigh processes like the

convective lofting of ice particles and their subsequent sublimation must exist.

Within the framework of the second SPARC water vapour assessment, we present here a comprehensive comparison of δD data sets obtained from satellite observation. These data sets are evaluated from the upper troposphere to the lower mesosphere with a clear focus on the stratosphere. In the comparison, we focus on satellite observations made since the new

millennium. In that time, three satellite instruments have provided information on stratospheric δD: Odin/SMR, Envisat/MIPAS, and SCISAT/ACE-FTS. The satellites were launched in 2001, 2002, and 2003, respectively. While Envisat ceased its operation in 2012, the other two instruments are still performing observations. In an earlier work, Lossow et al. (2011) compared HDO data observed by these three instruments. They found a good

agreement in the MIPAS and ACE-FTS data sets in the stratosphere, while the SMR data set showed a low bias. This bias could be explained by uncertainties in the spectroscopic parameters. Among the data sets, a high degree of consistency in the latitudinal distribution of HDO was found. In terms of δD no such comparisons exist. Instead, the data sets were analysed individually, typically focusing on the variability of dehydration in the tropical

tropopause layer and lower stratosphere. Nassar et al. (2007) used ACE-FTS observations from 2004 and 2005 to examine the dehydration in TTL. They found δD values between −700 per mille and −600 per mille and seasonal variation that is more obvious in the Northern Hemisphere. Steinwagner et al. (2010) expanded on this and showed a tape recorder signal in



δD using MIPAS observations from 2002 to 2004, corroborating the dominant role of the slow ascent for the transport of water vapour from the troposphere to the stratosphere. Later, Randel et al. (2012) evaluated ACE-FTS data from 2004 to 2009 in this regard and found a coherent tape recorder signal only up to about 20 km. This observational discrepancy remains

unsolved.

In this manuscript, we present coincident profile-to-profile and climatological comparisons of δD. For a better attribution and discussion of the issues in the isotopic ratio we also show the corresponding HDO and $H_2O$ results. In the next section we describe in detail the individual data sets. In Sect. 3 our approach for the profile-to-profile and climatological is outlined. In

Sect. 4 we present the results which will be summarised and discussed in Sect. 5.

## 2   Data sets

### 2.1   Odin/SMR

Odin is a Swedish-led satellite that is dedicated to both aeronomy and astronomy

observations. Launched on 20 February 2001, it uses a sun-synchronous orbit with equator-crossing times of about 6 LT and 18 LT on the descending and ascending node, respectively. Two instruments are deployed aboard the satellite; one of them is the Sub-Millimetre Radiometer (SMR). It measures the thermal emission at the atmospheric limb using a 1.1 m telescope. The instrument consists of five radiometers that cover several frequency bands

between 486 GHz and 581 GHz and around 119 GHz (Frisk et al., 2003). For the detection of the measured signal, either one acousto-optical spectrograph or one of two autocorrelators are used. The SMR observations typically cover the latitude range between 82.5°S and 82.5°N based on measurements along the orbital track. Since 2004, measurements off the orbital track have been performed during some seasons to obtain full latitudinal coverage from pole to

pole. This particularly concerns the boreal winter time.

The HDO and $H_2O$ information used in the present work is retrieved from emission lines centred at 490.597 GHz and 488.491 GHz, respectively (Urban et al., 2007). These lines are always measured with the 495A2 radiometer that can be tuned to measure a maximum bandwidth of 0.8 GHz within a frequency range from 486 GHz to 504 GHz. As the spectral

separation of the two emission lines is larger than the maximum bandwidth, HDO and $H_2O$





information cannot be obtained at the same time; consequently, no δD information on a single profile basis is available. Typically the HDO and $H_2O$ observations are performed in an alternating manner, with one orbit covering HDO, followed by one orbit covering the $H_2O$ emission line. Since SMR has a multitude of measurement targets and modes, the HDO and

$H_2O$ bands considered here are not observed on a daily basis. Until 25 April 2007, the typical observation frequency was 3–4 days per month. After that, the astronomy observations ceased, leading to an increased observation frequency of 8–9 days a month. The band is typically observed in the altitude range between 7 and 110 km with an effective vertical sampling of 3 km. Such scan takes about 140 seconds, which corresponds to a horizontal

sampling of 1 scan per 1000 km.

In the comparisons, we consider data that was derived at the Chalmers University of Technology in Gothenburg, Sweden using retrieval version 2.1 (Jones et al., 2009; Lossow et al., 2011). The retrieval of HDO and $H_2O$ profiles is based on the Optimal Estimation Method (OEM, Rodgers, 2000) using spectroscopic data from the Verdandi database (Eriksson, 1999).

Both HDO and $H_2O$ information can be retrieved in the altitude range between 20 and 70 km with a vertical resolution that is very close to the vertical sampling (Urban et al., 2007). The precision of a single HDO scan is best at around 30 km with about 20 %. Towards the lower and upper boundaries, the precision degrades to values beyond 50 % on average. For $H_2O$, the single-scan precision is for a large part of the covered altitude range better than 10 %. At the

profile boundaries, the precision is typically in the order of 30 %.

Earlier comparisons have shown that the SMR HDO data exhibit a dry bias in the stratosphere (Lossow et al., 2011). A dry bias has also been observed in $H_2O$ in the upper stratosphere and lower mesosphere (Hegglin et al., 2013). In this altitude region, a positive drift is currently being investigated.

The SMR data are screened according to three parameters, namely the retrieval quality flag, the measurement response, and a $\chi^2$ flag. The quality flag marks if a retrieved profile shall be used in any scientific analysis based on a number of retrieval diagnostics. Here we considered only data with the two best quality categories, i.e. 0 and 4. The measurement response describes the relative contribution of observational and a priori information to the retrieved

data. We chose a minimum measurement response of 70 % to limit the influence of the a priori data on our results. The $\chi^2$ flag is an additional criterion to earlier studies aiming to improve the screening of unreasonable data. This flag combines the fit quality of the retrieved



values towards the measurement and that of the true state towards the a priori. In general, the first part contributes the most. For HDO, only data with a $\chi^2$ flag ranging from 0.03 and 0.8 have been taken into consideration; for $H_2O$, data in the range between 0.03 and 0.6 have been considered. Application of this criterion rejects about 3 % of the HDO profiles and 4 %

of the $H_2O$ profiles. Scientifically usable data dates back to November 2001. Earlier data are not recommended due to problems with the instrument pointing. After consultation with the SMR data team, we only included data until May 2009 in the comparisons, due to a combination of calibrational issues, instrument drifts, and also some processing issues. In general, only limited HDO data are currently available after April 2011 due to a frequency

drift, typically less than 10 %, on a yearly basis relative to number of available profiles in 2008. This drift is basically handled by the retrieval scheme; however, larger frequency shifts trigger indirect forward model problems that cause an incorrect representation of the line shape. In total, 77,000 HDO profiles and 83,000 $H_2O$ profiles are available for the comparisons presented here.

## 2.2  Envisat/MIPAS

The Michelson Interferometer for Passive Atmospheric Sounding (MIPAS) was a cooled high-resolution Fourier transform spectrometer aboard Envisat (Environmental Satellite). This satellite was launched on 1 March 2002 and performed observations until 8 April 2012, when

communication with it broke down. Envisat orbited the Earth 14 times a day on a polar, sun-synchronous orbit with an altitude of about 790 km. The equator crossing times were 10 and 22 LT for the descending and ascending node, respectively. MIPAS measured thermal emission at the atmospheric limb in the wavelength range between 4.1 and 14.6 μm (685–2410 cm$^{-1}$) and covering all latitudes. MIPAS observations from July 2002 to March 2004,

which is referred to as the full resolution period of MIPAS, were included in the comparison. During this period, measurements used a spectral resolution of 0.035 cm$^{-1}$ (un-apodised) and covered the altitude range between 6 and 68 km, with a vertical sampling of 3 (up to 42 km) to 8 km. A whole scan took 76 s, corresponding to a horizontal sampling of 1 scan per 530 km. Later MIPAS observations used a coarser spectral resolution due to an instrument

problem, and also the vertical sampling pattern was changed (Fischer et al., 2008).

We utilise two sets of HDO and $H_2O$ data in the comparisons, namely retrieval version 5 and





20. Both sets are retrieved with the IMK/IAA processor, which has been developed in cooperation by the "Institut für Meteorologie und Klimaforschung" (IMK) in Karlsruhe (Germany) and the "Instituto de Astrofísica de Andalucía" (IAA) in Granada (Spain). The retrieval approach for both data sets is the same; the only difference is the calibration of the spectra provided by the European Space Agency (ESA). Retrieval version 5 is based on calibration version 3 (Lossow et al., 2011), while retrieval version 20 uses calibration version 5. The retrieval of HDO and $H_2O$ used spectral information from 14 microwindows located between 6.7 and 8.0 μm (1250–1483 cm$^{-1}$) (Steinwagner et al., 2010). It should be noted that this $H_2O$ data set is different from the nominal IMK/IAA $H_2O$ data set (see studies by Schieferdecker et al., 2015b or Lossow et al., 2017 for example) which is based on spectral information of 12 microwindows between 6.3 and 12.6 μm (796–1579 cm$^{-1}$). As the SMR retrieval, the MIPAS retrieval uses a non-linear least square approach. To avoid unphysical oscillations in the retrieved profiles, a first-order Tikhonov-type regularisation (Tikhonov and Arsenin, 1977) is employed. Spectroscopic data are taken from a compilation set up especially for the MIPAS mission (Flaud et al., 2003) which considers for this retrieval data from the updated version of HITRAN-2000 (High Resolution Transmission, Rothman et al., 2003). HDO information can be retrieved between about 10 and 60 km with a typical single scan precision of 20 % below 50 km and around 100 % at 60 km. In the upper troposphere and lower stratosphere, the vertical resolution is typically about 5 km. Towards higher altitudes, the resolution degrades and in the upper stratosphere and lower mesosphere it is in the order of 8 to 10 km. For $H_2O$, the vertical range is extended at the top. The single scan precision is in general within 10 %. The vertical resolution is worse than for HDO up to the middle stratosphere, while it is better in the upper stratosphere and lower mesosphere.

The MIPAS data are screened according to the visibility flag and the averaging kernel diagonal criterion. Data below the lowermost useful tangent altitude are identified by the visibility flag (set to zero in this case). This lowermost altitude is often determined by clouds, whose inference is derived using a cloud index (Spang et al., 2004). Clouds are rigorously screened, resulting in a bias towards cloud-free situations. The averaging kernel diagonal indicates the measurement contribution to the retrieved data, similar to the measurement response used to screen the SMR data. Data with a diagonal value of less than 0.03 are discarded to ensure at least a minimum of measurement information. In addition, data above the uppermost tangent altitude are not considered any further. However, the impact on this work is negligible. Overall, the v5 data set consists of more than 460,000 simultaneous



observations of HDO and $H_2O$ with a nearly daily coverage. More than 480,000 profiles comprise the v20 data set.

### 2.3    SCISAT/ACE-FTS

ACE-FTS (Atmospheric Chemistry Experiment - Fourier Transform Spectrometer) is one of three instruments aboard the Canadian SCISAT (or SCISAT-1) satellite (Bernath et al., 2005). It was launched on 12 August 2003 into a high inclination orbit with an altitude of 650 km. This orbit provides a latitudinal coverage from 85°S to 85°N, but is optimised for observations at high and mid-latitudes. Like MIPAS, ACE-FTS performs observations in the
infrared. The instrument covers the wavelength range between 2.3 and 13.3 μm (750–4400 $cm^{-1}$) with a high spectral resolution of 0.02 $cm^{-1}$. The observations are based on the solar occultation technique. ACE-FTS scans the Earth's atmosphere 30 times a day (15 sunrises and 15 sunsets) from about 5 to 150 km. The vertical sampling varies with altitude, ranging from about 1 km in the middle troposphere, to 3 to 4 km at around 20 km, and 6 km in the upper
stratosphere and mesosphere.

The ACE-FTS retrieval is based on an unconstrained, non-linear, least-squares, global-fitting technique (Boone et al., 2005, 2013) In the comparisons we considered data from two retrieval versions, i.e. from the well-validated version 2.2 and the newer version 3.5. The HDO retrieval of version 2.2 (Nassar et al., 2007) is based on spectral information from 24
microwindows in two separated wavelength intervals. One interval ranges from 3.7 to 3.8 μm (2612– 2673 $cm^{-1}$) and the other ranges from 6.7 to 7.1 μm (1402 –1498 $cm^{-1}$). This yields HDO profiles covering altitudes from 5.5 to 37.5 km. The single-scan precision is in general better than 10 % below 30 km. At the top of the profiles, the precision typically amounts to 25 %. In retrieval version 3.5, the number of microwindows was increased to 26. In addition, the
2 wavelength intervals were extended, i.e. they range from 3.7 to 4.0 μm (2493–2673 $cm^{-1}$) and from 6.6 to 7.2 μm (1383–1511 $cm^{-1}$). As a result of these changes, HDO information can be retrieved at higher altitudes, typically up to 49.5 km. The single-scan precision is rather similar to version 2.2. At 49.5 km, the precision is about 50 %. The $H_2O$ retrieval of version 2.2 uses 60 microwindows, providing profile information up to 89.5 km (Carleer et al., 2008).
As for HDO retrieval, the microwindows are separated in 2 wavelength intervals, ranging from 5.0 to 7.3 μm (1362 –2000 $cm^{-1}$) and from 10.3 to 10.5 μm (953–974 $cm^{-1}$). In version



3.5, the microwindows were optimised once more. Using 54 microwindows between 3.3 and 10.7 μm (937–2993 cm$^{-1}$), retrieval version 3.5 yields again an extension of the upper altitude limit where $H_2O$ information can be retrieved. The single-scan precision of $H_2O$ profiles is within 5 % for both retrieval versions. The spectroscopic data employed in the ACE-FTS

retrieval is taken from HITRAN-2004 (Rothman et al., 2005), including some more recent updates as detailed in Boone et al. (2013). The vertical resolution is the same for all data sets and corresponds to the vertical sampling outlined above. Like for MIPAS, δD can be derived on a single-profile basis.

The screening of the ACE-FTS data depends on the retrieval version. For version 2.2, the

"data issues page" (https://databace. scisat.ca/validation/data_issues_table.php) is taken into account, which lists problematic occultations. Besides the occultations flagged with "do not use", we also discard occultations that are marked with the label "use with caution". Data after September 2010 are not used due to problems with the input pressure and temperature data utilised in the retrieval. In version 3.5, these problems are resolved and data until the end of

2014 are used in this study. The screening of this data set uses the flag system described in detail by Sheese et al. (2015). We remove all profiles that are flagged with a value between 3 and 7 at any altitude. These numbers indicate either outliers, a lack of data to perform the outlier analysis, or instrument or processing errors. In total, 22,000 simultaneous observations of HDO and $H_2O$ for version 2.2 are available for comparison. For the longer, version 3.5

data set, there are slightly more than 40,000 profiles.

# 3  Approach

The present quality assessment of δD, HDO, and $H_2O$ data focuses primarily on the stratosphere. As a complement, we also show data in the upper troposphere and lower

mesosphere. The comparisons use different approaches. They are based on coincident measurements as well as monthly and seasonal averages.

A set of simultaneous HDO and $H_2O$ observations can be in general combined to a δD product, as for example an average, in two different ways:

(1) Calculate δD based on the individual HDO and $H_2O$ observations and subsequently

combine δD to the product in question. Here we denote this approach as "individual"



approach.

(2) Derive first the product in question separately for HDO and $H_2O$ and combine them subsequently to $\delta D$. Here we refer to this approach as "separate" approach.

In general, these two approaches are not commutative in a mathematical sense and will yield different results. While we prefer the individual approach, the results presented in this work are based on the separate approach. The main motivation for this is consistency, as the SMR observations do not allow the derivation of $\delta D$ on a single-profile basis (see Sect. 2.1). In the supplement to this manuscript, we show $\delta D$ results from MIPAS and ACE-FTS that are based on the individual approach. Those results are also compared to the results that we show here in the main part of the manuscript to provide estimates of the differences caused by the different approaches. For the comparisons, the profiles of the individual data sets are interpolated on a common regular pressure grid, consisting of 32 levels per pressure decade and ranging from 421 to 0.1 hPa. For MIPAS and ACE-FTS, a consistent set of HDO and $H_2O$ observations is used. If an HDO observation is not available (due to a failed retrieval or screening) then the simultaneous $H_2O$ profile is not used in the comparisons and vice versa.

To handle data that could potentially negatively influence the comparison results, we defined further screening criteria in addition to the standard screening of the individual data sets outlined in Sect. 2. The screening criteria are based on intervals for the volume mixing ratios of HDO and $H_2O$ and the $\delta D$ isotopic ratio. The intervals are listed in Tab. 1. They are fairly large, targeting the most obvious outliers that passed the previous screenings. Profiles that exhibit data points outside these intervals are discarded. For HDO and $H_2O$, this typically concerns only a handful of profiles. Solely, for the ACE-FTS v2.2 data set, the number of affected profiles is higher, i.e. 30 for HDO and 50 for $H_2O$. Considerably more profiles are screened for $\delta D$. The percentage ranges from around 0.5 % for the MIPAS data sets to 1 % for ACE-FTS v3.5 and 1.8 % for ACE-FTS v2.2. Note that HDO and $H_2O$ can have negative volume mixing ratios due to measurement noise that propagates through the retrieval. These volume mixing ratios are not removed; however, in combination they can cause isotopic ratios below the theoretical limit of $-1000$ per mille.

In the following, we describe the approaches and considerations for the different comparisons. First we take the profile-to-profile comparisons, then the seasonal comparisons, and finally the monthly comparisons.



### 3.1 Profile-to-profile comparisons

#### 3.1.1 Coincidences

In the profile-to-profile comparisons, we consider in general observations from two data sets
as coincident when they meet the following criteria: (1) a spatial separation of less than 1000
km, (2) a temporal separation that does not exceed 24 h and a separation of less than 5° both
in (3) geographical and (4) equivalent latitude. In comparisons between the two MIPAS data
sets or the two ACE-FTS data sets, the exact same observations are used (see Tab. 2). For the
equivalent latitude criterion, we use a stratospheric average value derived from MERRA
(Modern Era Retrospective-Analysis for Research and Applications, Rienecker et al., 2011)
reanalysis data. In cases where multiple coincidences are found for one observation, the
coincidence closest in space is used, given the small local time variation of stratospheric
water vapour (Haefele et al., 2008). In the troposphere and higher up in the mesosphere, this
variation can be substantial; however, these regions are not the main focus of this study.
Additionally, we consider only unique coincidences, i.e. once an observation is considered as
a coincidence it is not used any further as coincidence for other observations.

#### 3.1.2 Consideration of differences in the vertical resolution

The vertical resolution of the different HDO and $H_2O$ data sets varies as described in Sect. 2.
In particular, the MIPAS HDO and $H_2O$ data sets have a lower vertical resolution compared
to the other data sets. These differences only require a consideration in the comparisons at
altitudes where the vertical distribution of the parameter in question is highly structured. This
especially concerns the hygropause region in the lowermost stratosphere. The maximum of
HDO and $H_2O$ around the stratopause is relatively broad. This makes differences in the
vertical resolution a smaller issue than for the hygropause region, yet a direct comparison may
still not be appropriate.

The differences in the vertical resolution in the HDO and $H_2O$ comparisons to MIPAS are
accounted for by the method of Connor et al. (1994). Using the averaging kernel A and the a
priori profile $x_{apriori}$ from the MIPAS retrieval, the higher vertically resolved SMR and ACE-
FTS data ($x_{high}$) can be degraded onto the MIPAS vertical resolution as follows:





$$x_{deg} = x_{apriori} + A \cdot (x_{high} - x_{apriori}) \tag{3}$$

The degraded data $x_{deg}$ can subsequently be compared directly to the lower resolved MIPAS data. The comparisons between SMR and ACE-FTS are performed directly, as these data sets have a very similar vertical resolution in the altitude range where they overlap.

Also, $\delta D$ shows pronounced structures in its vertical distribution. Most relevant for this work is the structure in the tropopause region (Webster and Haymsfield, 2003; Nassar et al., 2007). Thus, comparisons of individual $\delta D$ profiles between the MIPAS and ACE-FTS data sets (see Supplement) should again consider the differences in the vertical resolution. However, the approach described by Eq. (3) cannot be applied in this case. $\delta D$ is here not a retrieved

quantity and as such no averaging kernels exist.

### 3.1.3   Bias determination

The bias $\bar{b}(P, z)$ between two coincident data sets for a given parameter P (i.e. HDO, $H_2O$, or $\delta D$) at an altitude z is derived as follows:

$$\bar{b}(P, z) = \frac{1}{n_c(P,z)} \cdot \sum_{i=1}^{n_c(P,z)} b_i(P, z) \tag{4}$$

In the equation, $n_c(P, z)$ represents the number of coincident measurements and $b_i(P, z)$ denotes the individual differences between those. We have considered these differences in absolute terms:

$$b_i(P, z) = b_{i,abs}(P, z) = x_i(P, z)_1 - x_i(P, z)_2 \tag{5}$$

and also in relative terms:

$$b_i(P, z) = b_{i,rel}(P, z) = \frac{x_i(P,z)_1 - x_i(P,z)_2}{[x_i(P,z)_1 + x_i(P,z)_2]/2} . \tag{6}$$

$x_i(P, z)_1$ and $x_i(P, z)_2$ are the individual observations of the two data sets that are compared with each other. For relative bias, the average over both data sets is used as the reference. This follows the convenience argument employed by Randall et al. (2003), given the a priori

knowledge that the data sets have large uncertainties, especially for HDO and $\delta D$.

Before we derive the overall bias $\bar{b}(P, z)$, we perform an additional screening on individual





biases $b_i(P, z)$ that are obvious outliers and would skew the bias estimates. This screening employs the median and median absolute deviation (MAD; e.g. Jones et al., 2012), which are more robust quantities in terms of larger outliers. We discarded individual biases that were outside the following interval: $\langle \text{median}[b_i(P, z)] \pm 10 \cdot \text{MAD}[b_i(P, z)] \rangle$ with $i = $

$1, \ldots, n_c(P, z)$. A factor of 10 for the median absolute deviation corresponds to a factor of about 7.5 for the standard deviation considering a normally distributed set of data. Thus the screening is relatively weak and aims to detect the most prominent outliers.

For the δD bias, the terminology above applies only for the individual approach, for which results are shown in the Supplement. In the separate framework, we calculate first average

profiles of HDO and $H_2O$ (again described by P) for each data set separately from the coincident set of data:

$$\bar{x}(P, z) = \frac{1}{n_c(P,z)} \cdot \sum_{i=1}^{n_c(P,z)} x_i(P, z). \tag{7}$$

In this step we consider exactly the same data points of the data sets that are compared for a given parameter, i.e. if a data point does not exist in one data set (due to missing coverage or

screening) the corresponding data point in the other data set is also not considered. The average HDO and $H_2O$ profiles of a given data set are subsequently combined to an average δD profile, following Eq. (1):

$$\bar{x}(\partial D, z) = \left[ \frac{\bar{x}(HDO, z)}{2 \cdot VSMOW \cdot \bar{x}(H_2O, z)} - 1 \right] \cdot 1000. \tag{8}$$

The absolute δD bias between two coincident data sets is then calculated as:

$$\bar{b}(\partial D, z) = \bar{x}(\partial D, z)_1 - \bar{x}(\partial D, z)_2. \tag{9}$$

For relative bias, we follow the approach outlined in Eq. (6):

$$\bar{b}(\partial D, z) = \frac{\bar{x}(\partial D, z)_1 - \bar{x}(\partial D, z)_2}{[\bar{x}(\partial D, z)_1 + \bar{x}(\partial D, z)_2]/2}. \tag{10}$$

### 3.1.4   More considerations

The results we present for the profile-to-profile comparisons are based on all available coincidences. No separation into specific seasons or latitude band has been considered, thus





neglecting these dependencies in Eq. (4) to Eq. (10). Furthermore, only biases that are based on at least 20 coincident observations are taken into consideration to avoid spurious results. This concerns mostly the lower and upper altitude boundaries where comparisons are possible.

## 3.2 Comparisons of seasonally averaged latitude cross sections

### 3.2.1 Data binning

To provide an overview of how the data sets compare as functions of time and space, we consider latitude cross sections for different seasons. Accordingly, we average all available

data for a given parameter for all seasons (i.e. MAM, JJA, SON, and DJF, represented by t) and latitude bins $\phi$ of $10°$ (centred at $85°S$, $75°S$, ..., $75°N$, and $85°N$):

$$\bar{s}(P, t, \phi, z) = \frac{1}{n_o(P,t,\phi,z)} \cdot \sum_{i=1}^{n_o(P,t,\phi,z)} x_i(P, t, \phi, z). \qquad (11)$$

Here, $n_o(P, t, \phi, z)$ describes the total number of observations $x_i(P, t, \phi, z)$ that fall into a specific bin. Before these data are averaged, they are screened using again the median and

median absolute difference. Here we use the interval $\langle \text{median}[x_i(P, t, \phi, z)] \pm 7.5 \cdot \text{MAD}[x_i(P, t, \phi, z)]\rangle$ with $i = 1, ..., n_o(P, t, \phi, z)$. This screening is a bit more strict than for the profile-to-profile comparisons, as typically less data fall into individual bins. Besides the average, we also calculate the corresponding standard error $\epsilon(P, t, \phi, z)$:

$$\epsilon(P, t, \phi, z) = \sqrt{\frac{1}{n_o(P,t,\phi,z) \cdot [n_o(P,t,\phi,z)-1]} \cdot \sum_{i=1}^{n_o(P,t,\phi,z)} [x_i(P, t, \phi, z) - \bar{s}(P, t, \phi, z)]^2}. \qquad (12)$$

Averages that are based on fewer than 20 individual observations or are less than its associated standard error in absolute terms are not considered any further.

Within the separate framework, the seasonally averaged latitude cross sections for $\delta D$ are calculated from the corresponding results of HDO and $H_2O$:

$$\bar{s}(\partial D, t, \phi, z) = \left[\frac{\bar{s}(HDO,t,\phi,z)}{2 \cdot VSMOW \cdot \bar{s}(H_2O,t,\phi,z)} - 1\right] \cdot 1000. \qquad (13)$$

The corresponding standard error is calculated according to the Gaussian error propagation:





$$\epsilon(\partial D, t, \phi, z) = \frac{500}{VSMOW} \cdot \sqrt{\left[\frac{\epsilon(HDO,t,\phi,z)}{\bar{s}(H_2O,t,\phi,z)}\right]^2 + \left[\frac{\bar{s}(HDO,t,\phi,z) \cdot \epsilon(H_2O,t,\phi,z)}{\bar{s}(H_2O,t,\phi,z)}\right]^2} . \tag{14}$$

Given that this information is not available from all data sets, this calculation assumes no error covariance between the HDO and $H_2O$ data to keep consistency.

### 3.2.2 Summarising the results

For a summary of these comparisons, we provide some additional quantities by combining the results from all individual latitude bands. Results from at least 6 latitude bands are required, otherwise the combined quantity is discarded. As a first quantity, we consider the average bias $\bar{b}_\phi(P, t, z)$ of the latitude cross sections from two data sets:

$$\bar{b}_\phi(P, t, z) = \frac{1}{n_\phi(P,t,z)} \cdot \sum_{i=1}^{n_\phi(P,t,z)} [\bar{s}(P, t, \phi_i z)_1 - \bar{s}(P, t, \phi_i z)_2]. \tag{15}$$

The subscripts at the end of the variables again refer to the two data sets. Besides the average, we look also at the de-biased standard deviation:

$$\sigma_\phi(P, t, z) = \sqrt{\frac{1}{n_\phi(P,t,z)-1} \cdot \sum_{i=1}^{n_\phi(P,t,z)} [\bar{s}(P, t, \phi_i, z)_1 - \bar{s}(P, t, \phi_i, z)_2 - \bar{b}_\phi(P, t, z)]^2} . \tag{16}$$

The de-biased standard deviation is generally interpreted as measure of the combined precision of the two data sets that are compared (von Clarmann, 2006). In this specific case, it describes the combined precision of the seasonally averaged latitude cross-section from two data sets. As a last step, we consider the correlation coefficient $r_\phi(P, t, z)$ between the latitudinal cross sections from two data sets:

$$r_\phi(P, t, z) = \frac{\sum_{i=1}^{n_\phi(P,t,z)}[\bar{s}(P,t,\phi_i,z)_1 - \bar{s}_\phi(P,t,z)_1] \cdot [\bar{s}(P,t,\phi_i,z)_2 - \bar{s}_\phi(P,t,z)_2]}{\sqrt{\sum_{i=1}^{n_\phi(P,t,z)}[\bar{s}(P,t,\phi_i,z)_1 - \bar{s}_\phi(P,t,z)_1]^2} \cdot \sqrt{\sum_{i=1}^{n_\phi(P,t,z)}[\bar{s}(P,t,\phi_i,z)_2 - \bar{s}_\phi(P,t,z)_2]^2}} \tag{17}$$

with

$$\bar{s}_\phi(P, t, z)_1 = \frac{1}{n_\phi(P,t,z)} \cdot \sum_{i=1}^{n_\phi(P,t,z)} \bar{s}(P, t, \phi_i z)_1 \text{ and} \tag{18}$$

$$\bar{s}_\phi(P, t, z)_{21} = \frac{1}{n_\phi(P,t,z)} \cdot \sum_{i=1}^{n_\phi(P,t,z)} \bar{s}(P, t, \phi_i z)_2, \tag{19}$$





which is the average of the seasonally averaged latitude cross-section over all latitude bins. No error estimates are considered in the calculation of the correlation coefficient, since the latitudinal cross sections from two data sets are expected to be highly correlated and not simply correlate by chance.

### 3.2.3 More considerations

These comparisons do not take into account the differences in the vertical resolution between MIPAS and the other data sets. Obtaining the convolution data to adapt the vertical resolution of non-coincident data sets is not a trivial procedure. Risi et al. (2012) and Schieferdecker

10 (2015a) describe methods of doing so. Since data averaging tends to reduce differences in the vertical resolution, we decided not to consider this aspect any further.

## 3.3 Comparisons of monthly averaged profiles in the tropics

In the final comparison we consider the tropics, a key region in particular for lower
15 stratospheric water vapour. We focus on February, April, August, and October which are the months the ACE-FTS observations typically provide tropical coverage. In recent years, there has also been some very limited coverage in May and November due to the shift of the SCISAT orbit, but these months are not considered here. The calculation of the monthly averaged profiles follows the same scheme as outlined by Eq. (11) to Eq. (14), except that
20 only one latitude bin (15°S – 15°N) is considered and that the seasonal dependence is replaced by a monthly dependence. Screening before and after the average calculation is the same as used for the seasonally averaged latitude cross sections (see Sect. 3.2). Again, no degradation of the ACE-FTS and SMR data onto the vertical resolution of the MIPAS data set is considered.



# 4 Results

## 4.1 Profile-to-profile comparison

In Fig. 1, the results of the profile-to-profile comparisons for δD (upper row), HDO (middle row) and $H_2O$ (bottom row) are shown, derived according to the description in Sect. 3.1. In

the left panels the absolute biases are shown, while the right panels show the relative biases. Tab. 2 summarises the characteristics of the comparisons between the different data sets in terms of their overlap period, covered latitude range, number of coincidences, and the average separation in time and space.

The comparisons are typically based on several thousand coincidences and cover almost all

latitudes and a time period of multiple years. The only exception is the comparisons between the MIPAS and ACE-FTS data sets. Given that the MIPAS data sets end in March 2004 and the ACE-FTS observations effectively start in the second half of February 2004, there is only a very limited overlap period. During the short overlap period, the majority of ACE-FTS observations occurred in March at northern polar latitudes. Overall, these comparisons cover

only latitudes from 51°N to 83°N, and most of the coincidences are concentrated near 70 N. The number of coincident profiles vary between 300 and 400, depending upon which of the MIPAS and ACE-FTS data sets are compared with each other. Also, the average separation in time and longitude are larger for the comparisons between the MIPAS and ACE-FTS data sets than for any of the other comparisons.

The largest deviations in δD are found in the comparisons to the SMR data set. Between the SMR and MIPAS data sets, the absolute bias ranges from –230 per mille near the 50 hPa region to almost 150 per mille in the 1 hPa region. In relative terms, the corresponding numbers are 40 % at 100 hPa and –40 % in the stratopause region. The biases change signs slightly below 4 hPa. The comparisons of the SMR data set with the coincident ACE-FTS

profiles show similar biases to the ones described above, with a peak deviation of –250 per mille around 60 hPa. The biases decrease in size to –100 per mille at 20 hPa. Above 15 hPa, the biases to the ACE-FTS v3.5 data set are smaller than the bias to the ACE-FTS v2.2 data set. For example, at 4 hPa the biases amount to –55 per mille (15 %) and –30 per mille (7 %), respectively. Higher up, the ACE-FTS retrievals come closer to their upper limit and the

biases to the SMR data set exhibit distinct variations. The four comparisons between the different MIPAS vs. ACE-FTS data sets show in general a good agreement in δD. The biases





are typically within ±30 per mille between about 100 hPa and 4 hPa, corresponding to biases within ±10 % in relative terms. Below and above this altitude range, the biases deteriorate. Comparisons between data sets from the same instrument typically exhibit very small biases. Exceptions occur primarily towards the upper and lower boundaries of the data sets.

Close to 60 hPa, the HDO comparisons to the SMR data set exhibit biases of about –0.4 to – 0.3 ppbv (–125 % to –90 %). Towards higher altitudes the biases decrease in size; in the altitude range between 10 and 0.1 hPa, the biases are within ±25 %. The comparisons between the ACE-FTS and MIPAS data sets show a good agreement with deviations in the range of ±10 % in the lower and middle stratosphere. Biases with a larger size are only observed below

100 hPa and above 4 hPa. Also, for $H_2O$, the comparisons indicate very good agreement between the different MIPAS and ACE-FTS data sets (typically within ±10 %). The comparisons to the SMR data set show the best agreement between about 50 hPa and 10 hPa, where the biases to the MIPAS and ACE-FTS data sets do not exceed –0.5 ppmv (–10 %). Here, the biases typically have a larger size in the comparison to the MIPAS data sets than the

ACE-FTS data sets. Higher up, the deviations increase to about –25 %. In volume mixing ratio terms, this corresponds to biases between –2 and –1 ppmv. Again, the size of the deviations is larger in the comparisons to the MIPAS data sets.

### 4.2    Comparisons of seasonally averaged latitude cross sections

In this sections, the comparison results from the seasonally averaged latitude cross sections are presented. They aim to provide an overview of how the data sets compare as function of time and space. We start with three examples, focusing on cross sections at 100 hPa, 10 hPa and 1 hPa (Figs. 2 to 4). After that a summary of the results is presented (Figs. 5 to 7), as described in Sect. 3.2.2.

#### 4.2.1    Examples

#### 100 hPa

Figure 2 shows the latitudinal cross sections of δD (left column), HDO (centre column), and $H_2O$ (right column) for the 100 hPa level. The different rows consider the results for the

different seasons MAM, JJA, SON, and DJF. Dashed lines indicate the standard error of the





seasonal averages (see Eq. 12 and 14). 100 hPa corresponds to the level of the tropical tropopause where the temperature largely determines the transport rate of water vapour into the stratosphere. It is therefore an interesting altitude for comparisons. Both the MIPAS and ACE-FTS data sets cover this altitude, even though it is not far away from the lower limit of

possible retrievals for both instruments. The SMR observations provide only HDO results at this altitude and therefore are not shown in this comparison.

The ACE-FTS data sets show typically the highest depletion in the tropics. The v3.5 data set indicates consistently lower $\delta D$ values than the v2.2 data set. Towards higher latitudes, the $\delta D$ values generally increase. A pronounced decrease is observed in the Antarctic in JJA and

SON. For the MIPAS data sets, the latitudinal structure in MAM resembles a "W", with a maximum in the inner tropics and minima at subtropical latitudes. This structure mainly originates from a pronounced deviation in $H_2O$. Polewards of 60° latitude, the $\delta D$ values vary less. A similar picture is visible in DJF. Unlike in the ACE-FTS data sets, the MIPAS data sets do not show any decrease in the Antarctic in JJA and SON. In both seasons, the MIPAS

data sets indicate a maximum in the northern tropics and lower $\delta D$ values towards the Arctic. In JJA, the $\delta D$ values derived in the high Antarctic actually indicate the lowest depletion compared to all other latitudes. Overall, deviations of 100 per mille and beyond are observed occasionally. For HDO, the deviations are visually smaller. The latitude dependence of the data sets is more coherent than for $\delta D$, with some obvious exceptions observed both in the

tropics and the higher latitudes. In JJA and SON, the ACE-FTS data sets show a distinct drop in the volume mixing ratios polewards of 60°S as well, which is not captured by the MIPAS data sets. Also, latitude dependence is more consistent among the data sets for $H_2O$ than for $\delta D$ in general. The best agreement between the MIPAS and ACE-FTS data sets is found in SON. In particular, the MIPAS v20 data set shows a pronounced deviation at 30°S and 30°N

in MAM and DJF. Like spikes, the amount of water vapour increases prominently by 0.5–1 ppmv in a small latitude range. This clearly influences the latitude dependence observed in $\delta D$, resulting in the "W"-shaped latitude dependence described above. A similar, but weaker, behaviour can be observed in the other seasons because the agreement with the other data sets is better, making it less obvious. In DJF, the MIPAS v5 data set shows a pronounced

deviation from the ACE-FTS data sets in $H_2O$ in the tropics and subtropics. The differences observed in the Antarctic in JJA and SON for $\delta D$ and HDO can also be observed in $H_2O$, but are less pronounced. Be reminded that the MIPAS $H_2O$ data sets included here are specially





retrieved versions with a lower vertical resolution adjusted to the HDO resolution, as described in Sect. 2.2.

**10 hPa**

At 10 hPa (Fig. 3), in the middle stratosphere, all three satellites provide data for comparison. The situation is clearly different from that at 100 hPa. The overall latitude dependence in δD is smoother and more evenly distributed compared to 100 hPa. The most pronounced difference in δD is that the SMR data set shows between 50 to 100 per mille more depletion than the MIPAS and ACE-FTS data sets, with little latitude dependence. In δD, the MIPAS
and ACE-FTS data sets exhibit smaller deviations in some regions. Consistently, for all seasons, the ACE-FTS v2.2 data set shows less depletion in δD than the ACE-FTS v3.5 and the two MIPAS data sets almost everywhere. The difference between the ACE-FTS v2.2 and the MIPAS data sets is typically about 25 per mille in the tropical regions, and often increasing somewhat towards the middle latitudes. Poleward of 60°S in SON the ACE-FTS
v2.2 and v3.5 data sets show a distinct increase from –450 to –350 per mille. A corresponding feature is also visible in both HDO with increases from 1 to 1.25 ppbv and $H_2O$ from 5.5 to 6.5 ppmv. This structure is not apparent in either the MIPAS or SMR data sets. For HDO, latitude dependence is very consistent among the data sets, as observed for δD. There is again a low bias in the SMR data set, which is approximately 0.15 ppbv and rather independent
from season and latitude. In $H_2O$, such a difference is not obvious. Instead, the SMR data set agrees quantitatively rather well with the ACE-FTS data sets. Notably, the MIPAS data sets typically exhibit the highest volume mixing ratios. Towards polar latitudes, this high bias often increases, occasionally exceeding 0.5 ppmv. In the tropics, the spread among the data sets varies between 0.2 and 0.5 ppmv, roughly.

**1 hPa**

In the stratopause region at 1 hPa (Fig. 4), the volume mixing ratio of water vapour in the middle atmosphere typically reaches its maximum. Data for comparison at this altitude are available from the SMR and MIPAS data sets. The ACE-FTS v3.5 data set has very limited
coverage of this altitude level and is thus omitted here. Similar to the previous altitudes, there



are pronounced deviations between the data sets. In δD, the sign of the bias between the SMR and MIPAS data sets is reversed compared to the 10 hPa level. The SMR data set shows typically 25 per mille to 100 per mille less depletion than the MIPAS data sets. The bias is more dependent on season than latitude. The largest deviations are visible in SON and DJF,

while in JJA the bias is smallest. In JJA and SON, there is a tendency for the bias to be largest in the Arctic. The MIPAS v20 data set shows consistently larger δD values than the v5 data set, but the deviation is on the order of only a few per mille. The cross sections exhibit rather similar structures in both HDO and $H_2O$, but the SMR observations show lower volume mixing ratios than the MIPAS data sets for all seasons and latitudes. In HDO, the deviation

varies between 0.1 and 0.2 ppbv and for $H_2O$, between 0.7 and 2.0 ppmv. While for HDO the biases show only a small seasonal dependence, the biases in $H_2O$ minimise in JJA. In general, the biases are a bit larger in the polar regions than in the tropics.

### 4.2.2 Summary of the seasonal comparisons

Figures 5 to 7 show summary plots for the comparisons of the seasonal cross sections (see Sect. 3.2.2) that were exemplarily described for three pressure levels in Fig. 2 to 4. The left column shows the biases between the data sets averaged over all latitudes, while the middle column shows the corresponding (de-biased) standard deviations. In the right column, the correlation coefficients between the cross sections are shown. Figure 5 presents the results for

δD, Fig. 6 those for HDO, and Fig. 7 the results for $H_2O$, respectively. Each figure shows the four different seasons in different rows. The summary figures extend the results obtained from the profile-to-profile comparisons shown in Fig. 1. This is particularly valuable for the comparisons between the MIPAS and ACE-FTS data sets, which were only possible with a very limited temporal and spatial overlap in the profile-to-profile approach.

**δD**

In general, the δD biases among the different data sets averaged over all latitudes yield very similar results to those derived from the profile-to-profile comparisons. Some prominent deviations are visible in the comparisons between the SMR and MIPAS data sets as well as in

the comparisons between those from MIPAS and ACE-FTS. For the latter, this concerns



primarily altitudes below 100 hPa. The profile-to-profile comparisons in Fig. 1 indicate a low bias of the MIPAS data sets, while here a clear high bias is found. For the comparison between the SMR and MIPAS data sets, the approaches yield differing results at 50 hPa as well as in the upper stratosphere and lower mesosphere. At about 50 hPa, the negative biases

are larger than in the profile-to-profile comparisons. The δD bias above 0.3 hPa in Fig. 1 approaches zero, while here the biases continue to increase for all seasons except for JJA. In general, a clear seasonal dependence is observed in the upper stratosphere and lower mesosphere for the comparison between the SMR and MIPAS data sets. At 0.6 hPa, the δD biases are roughly 75 per mille in MAM while in SON, DJF, and the profile-to-profile

comparisons they amount to 125–150 per mille. In JJA, the biases here are around 40 per mille. During this season, the bias between the SMR and MIPAS v5 data sets switches signs at about 0.3 hPa. Slightly below 0.1 hPa it amounts to about –50 per mille. This behaviour is not observed during other seasons. The de-biased standard deviations show a characteristic altitude dependence. Between 20 and 0.5 hPa, the deviations are typically smaller than 25 per

mille. The lowest values occur in comparisons among the data sets from the same instrument (typically less than 10 per mille). Below 20 hPa, the standard deviations increase significantly and occasionally exceed 100 per mille at altitudes lower than 100 hPa. Again, estimates for the comparisons among data sets from the same instrument are smaller, but they are still substantial. Above 0.5 hPa, the comparisons between the SMR and MIPAS data sets also

exhibit a pronounced increase in the de-biased standard deviations, maximising in MAM and SON. Values exceeding 100 per mille are also observed close to 0.1 hPa. The correlation coefficients reflect in many ways the picture found for the de-biased standard deviation. Very high correlation coefficients (close to 1) are found between about 20 and 2 hPa. Lower correlations are visible at altitudes where one of the data sets compared reaches one of its

vertical boundaries, like for the SMR data set around 50 hPa, the ACE-FTS v2.2 data set around 4 hPa (especially in SON), or for the ACE-FTS v3.5 data set close to 1 hPa. The comparisons between the MIPAS and ACE-FTS data sets show a distinct reduction of the correlations in a layer around 130 hPa, where the values turn negative in all seasons. In the lower mesosphere (above 0.3 hPa), the correlation coefficients between the SMR and MIPAS

data sets turn negative as well.





**HDO**

As for δD, the HDO bias results from the seasonal means averaged over all latitudes and the profile-to-profile comparisons are quite similar. Differences among these two approaches are visible again below 100 hPa concerning all comparisons. Higher up, above 1 hPa, the biases

between the SMR and MIPAS data sets are rather consistent with altitude in JJA, while during the other seasons and in the profile-to-profile comparisons they approach zero. The de-biased standard deviations paint a similar picture, qualitatively, to those from δD. Typically, the values in the middle and upper stratosphere are smaller than 0.07 ppbv. Below, the values increase again, in particular below 100 hPa (exceeding 0.5 ppbv). Also, in the lower

mesosphere, the comparisons between the SMR and MIPAS data sets exhibit increasing estimates. Close to 0.1 hPa, the standard deviations amount to almost 0.35 ppbv in MAM, 0.2 ppbv in SON, and around 0.15 ppbv in JJA and DJF. The correlation coefficients show the highest values also in the stratosphere, in particular in MAM, where values larger 0.9 are found between 80 and 1.5 hPa. In JJA and SON, a clear reduction (occasionally lower than

0.3) is found between 100 and 30 hPa in most comparisons (except between those data sets from the same instrument). This behaviour coincides with increases in the de-biased standard deviations.

**H₂O**

The bias results for $H_2O$ summarised in Fig. 7 resemble those from the profile-to-profile comparison shown in Fig. 1, similarly as for δD and HDO. Again, differences are visible below 100 hPa among the two comparison approaches. The comparisons between the MIPAS and ACE-FTS data sets exhibit larger biases around 2 hPa. They amount to more 0.5 ppmv, while they do not exceed 0.3 ppmv in the profile-to-profile comparisons. The biases relative

to the SMR data set show a seasonal variation around 1 hPa and the smallest estimates are observed in JJA. The de-biased standard deviations are typically smaller than 0.25 ppmv between 20 and 2 hPa. In this altitude range, the values are around 0.1 ppmv for the comparison among the different data sets from the same instrument. Lower down, the de-biased standard deviations increase again. In JJA and SON, a local maximum is observed

around 40 hPa, coinciding with a distinct local minimum in the correlation coefficients, which is also observed in the comparisons for HDO. In δD, this behaviour is not as obvious,



indicating some cancelation of the problems in HDO and $H_2O$. At 200 hPa, the de-biased standard deviations exceed 1 ppmv. In the lower mesosphere, the increase is more moderate (up to 0.7 ppmv). The highest correlations in the latitudinal distribution are found in about the same altitude region as $\delta D$ and HDO. A distinct local minimum (with negative values) is

observed in SON between 1 and 0.3 hPa. Similarly, a local minimum is found around 100 hPa in MAM and DJF. Be reminded again that the MIPAS $H_2O$ data sets are based on special retrievals, different to the nominal data sets (see Sect. 2.2).

### 4.3    Tropical monthly averaged profiles in the tropics

Figure 8 shows the tropical (15°S to 15°N) monthly mean profiles of $\delta D$ (left column), HDO (middle column), and $H_2O$ (right column) for February, April, August, and October (different rows). The ACE-FTS observations, which focus on middle and high latitudes, typically cover the tropics only during these months. For this reason, the monthly averaged data are preferable since they give a more appropriate picture of the isotopic ratio and the

corresponding water vapour profiles in this region instead of the seasonal averages used in the previous section. Our primary focus here is on the lower stratosphere. In this region, HDO and $H_2O$ exhibit the tape recorder signal manifesting itself in additional extrema in the vertical distribution above the hygropause. For $\delta D$, the existence of a tape recorder is under debate, as described in the Introduction.

**Lower stratosphere**

In February, both MIPAS data sets capture the tape recorder structure in $\delta D$ with the first minimum of –670 per mille near 100 hPa, followed by the maximum of –580 per mille propagating from last season at an altitude of 70 hPa. Above, the corresponding second

minima from previous winter of –600 per mille at 35 hPa can be seen. In the HDO and $H_2O$ data, however, the MIPAS data sets do not capture the tape recorder structure in the lower stratosphere. Instead, the minimum associated with the hygropause is rather wide, due to the rather low vertical resolution in the dry phase of the tape recorder. In $\delta D$ the ACE-FTS v3.5 shows a clear first minimum of –700 per mille at an altitude of approximately 100 hPa. This

minimum is wider in altitude than for the MIPAS data sets. Higher up there is no structure of




any tape recorder, except a small tendency to another minimum visible around 20 hPa. This is slightly more obvious for the ACE-FTS v2.2 data set, with a weak maximum at about 40 hPa and a second minimum close to 30 hPa. The first minimum around 100 hPa indicates approximately 50 per mille less depletion than for the v3.5 data set. This behaviour is

observed during all the months considered here. In contrast to the wide minimum observed in the MIPAS data sets, the ACE-FTS HDO and $H_2O$ data sets exhibit a distinct hygropause. A weak local maximum is found close to 50 hPa and a second minimum slightly above 30 hPa, both in HDO and $H_2O$. This structure is most pronounced in $H_2O$ for the v3.5 data set. The SMR $\delta D$ data deviate from the other datasets, in particular in terms of the absolute values.

The lower limit of this data set is close to 50 hPa. There, it seems to display the tape recorder maximum (around −700 per mille) and the second minimum is presumably located at 30 hPa (about −720 per mille). As seen in the previous comparisons, the SMR HDO data has a low bias of about 0.2 ppbv relative to the other datasets in the lower stratosphere. At the lower limit, the mixing ratio approaches zero and actually turns negative. However, the tape

recorder structure is well visible in these data. The altitudes of the maximum and second minimum are quite similar to those observed in the ACE-FTS data sets. In $H_2O$, these extrema are located slightly higher than in the ACE-FTS data.

In April, the first $\delta D$ minimum is clearly located higher in the MIPAS (around 75 hPa) than in the ACE-FTS data sets, which show this at about the same altitude as in February. The

MIPAS data sets show a rather constant isotopic ratio between 50 hPa and 10 hPa. Weak tape recorder structures are visible in the ACE-FTS data sets, but the extrema are located higher in altitude for the v3.5 data set. The SMR data set shows pronounced structures with a maximum slightly below 30 hPa and a minimum at 20 hPa. In HDO and $H_2O$, the tape recorder structures are clearly visible in all data sets with differences in the absolute mixing ratios and

the altitudes of the extrema. In particular, the maximum is located at a lower altitude in the MIPAS data sets. They show this at 50 hPa, while in the other data sets the maximum is located at about 40 hPa.

For $\delta D$, the situation in August is rather similar to that in April. In HDO, the deviations between the SMR and the other data sets are larger at the hygropause compared to April.

Also, the hygropause is higher in altitude in the ACE-FTS than the MIPAS data sets. Likewise, the tape recorder structures in $H_2O$ differ among the data sets. They are most pronounced in the SMR data set, but the maximum occurs at a lower altitude than in the other data sets. The second minimum has the highest location in the ACE-FTS data sets.



There is relatively good agreement among the data sets in the lower stratosphere with regard to HDO and $H_2O$ in October. Yet, pronounced differences in $\delta D$ between the MIPAS and ACE-FTS data sets are visible. The first minimum is observed at slightly below 100 hPa in the ACE-FTS data sets while it is located much higher, i.e. around 50 hPa, in the MIPAS data sets. Above, clear tape recorder structures are visible in all data sets from both instruments, but the shift in altitude remains.

**Altitudes above 10 hPa**

Focusing on altitudes above 10 hPa, the SMR data set shows lower depletion than the MIPAS data sets above about 3 hPa, as also observed in the seasonal comparisons.

The biases vary among in the individual months and maximise in the lower mesosphere in February. The SMR and ACE-FTS v3.5 data sets show a good agreement between 3 hPa and 1 hPa. Good agreement in HDO between the SMR and MIPAS data sets is observed in the altitude range from 1 to 0.2 hPa in February. Above 0.2 hPa, the SMR data set shows typically higher mixing ratios than the MIPAS data sets; elsewhere, the low bias already seen in the previous figure is visible. Between 7 and 1.5 hPa the MIPAS data sets show slightly higher HDO volume mixing ratios than the ACE-FTS data sets. At the upper limit of the ACE-FTS v3.5 data set, i.e. close to 0.9 hPa, it shows higher mixing ratios of 0.1 to 0.15 ppbv (largest in February and October) relative to the MIPAS data sets. In terms of $H_2O$, the data from the three instruments start to deviate above 10 hPa. The lowest mixing ratios are observed for the SMR data set. Up to about 0.5 hPa, the ACE-FTS data sets show lower mixing ratios than the MIPAS data sets (up to 0.75 ppmv), while above the picture is reversed. The altitude of the middle atmospheric water vapour maximum differs among the data sets from the three instruments. At 0.1 hPa, the MIPAS v5 data set shows a lower $H_2O$ mixing ratio than the v20 data set during all months. Likewise, the ACE-FTS v2.2 data set shows a lower amount of $H_2O$ than the v3.5 data set at this altitude.



## 5 Discussion

We have assessed the quality of satellite δD observations from the upper troposphere to the lower mesosphere using profile-to-profile and climatological comparisons. We find clear quantitative differences in the isotopic ratio. This concerns primarily the comparisons to the SMR data set. The MIPAS and ACE-FTS data sets agree rather well, with exceptions close to the vertical limit where observations can be made by these instruments, i.e. below 100 hPa and in the upper stratosphere.

In the profile-to-profile comparisons (Fig. 1), the SMR data set show a significantly higher depletion in δD than the MIPAS and ACE-FTS data sets in the lower stratosphere, which is close to its lower limit; correspondingly, the uncertainties are larger. The biases maximise close to 50 hPa and exceed -200 per mille. In the upper stratosphere and lower mesosphere, the picture is reversed, with the SMR data set instead showing less depletion than the MIPAS data sets. The biases switch signs close to 4 hPa. Above, the biases increase to almost 150 per mille in the lower mesosphere, where arguably the MIPAS data sets, close to their upper retrieval limit, also become more uncertain. Qualitatively, the same picture is found in the seasonal comparisons (averaged over all latitudes), with some quantitative variations among the seasons, in particular in the lower mesosphere. The δD biases of the SMR data set can be related to biases in both HDO and $H_2O$. For HDO, the SMR data set exhibits a low bias (Figs. 1 and 6) throughout all seasons. In the lower stratosphere, the biases are larger than 0.25 ppbv and then decrease with increasing altitude to be close to zero at 0.2 hPa. For $H_2O$, on the other hand, the SMR data set compares rather well with the MIPAS and ACE-FTS data sets in the lower stratosphere (Figs. 1 and 7), while in the upper stratosphere and lower mesosphere, distinct low biases are visible. The maximum $H_2O$ bias for the SMR comparisons in the upper stratosphere varies between 1 and 2 ppmv, depending on season. Overall, the δD biases of the SMR data set are driven by HDO in the lower stratosphere, while $H_2O$ is the clear driver in the upper stratosphere and lower mesosphere. In between, in the middle stratosphere, the biases in δD are a combination of deviations in both HDO and $H_2O$. Biases in both HDO and $H_2O$ exist around 4 hPa, where the δD biases of the SMR data set are close to zero compared to the MIPAS and ACE-FTS data sets, but they interfere favourably to reduce the δD bias.





### 5.1 SMR results

The bias in the SMR HDO data set has been attributed primarily to uncertainties in the spectroscopic parameters used in the retrieval (Lossow et al., 2011). Such uncertainties (they are larger than in the infrared region where MIPAS and ACE-FTS perform measurements)

might also contribute to the biases observed in $H_2O$; however, the dominant factor clearly arises from issues regarding the calibration. To a large extent, this is related to the sideband filtering affecting the retrievals. The SMR instrument is a heterodyne instrument measuring in a high-frequency range in the microwave region. To detect the measured signal $\nu_{RF}$ (radio frequency), it is converted down to a lower intermediate frequency $\nu_{IF}$ using a constant local

oscillator frequency $\nu_{LO}$ (Frisk et al., 2003). The relation between the three frequencies is described by Eq. (20)

$$\nu_{IF} = |\nu_{LO} - \nu_{RF}| \tag{20}$$

Due to this relationship, the intermediate frequency $\nu_{IF}$ holds the information from two frequency bands, i.e. the signal sideband and the image side band. For the $H_2O$ observations

the signal side bands range from 488.35 GHz to 488.75 GHz and from 488.95 GHz to 489.35 GHz. Given a local oscillator frequency $\nu_{LO}$ of 492.75 GHz, the corresponding image sidebands are located between 497.15 GHz and 496.75 GHz as well as between 496.55 GHz and 496.15 GHz. These image sidebands are suppressed with interferometers and in the retrievals, their leakage has been estimated to be 0.05 %. The latest leakage estimates vary

between less than 1 % and 5 % and exhibit some time dependence. The uncertainties in the sideband leakage and the calibration of the autocorrelator result in a positive drift of the SMR $H_2O$ results, mostly visible in the upper stratosphere and lower mesosphere (Khosrawi et al., 2018). Thus, over time, the low $H_2O$ biases in this altitude range decrease and likewise the biases in $\delta D$. This has an impact on the results from the profile-to-profile and seasonal

comparisons relative to MIPAS. While the profile-to-profile comparisons cover only the years from 2002 to 2004, the climatological comparisons include SMR data until 2009. In MAM and JJA in particular, reduced biases of $H_2O$ and $\delta D$ are visible in the upper stratosphere and lower mesosphere in the climatological comparisons relative to the profile-to-profile comparisons. There might be a small drift also in the SMR HDO data set, but it is not as

pronounced as in $H_2O$, and for HDO the main contribution to the bias will still originate from the spectroscopic parameters.



## 5.2 MIPAS and ACE-FTS results

The profile-to-profile comparisons (Fig.1) between the MIPAS and ACE-FTS data sets exhibit the best agreement in the 100 to 5 hPa region. Most of the time the MIPAS data sets

show a small high bias in δD, up to about 25 per mille (10 %). Also for the HDO and $H_2O$ data, the agreement in the same altitude region is rather good. But due to the short overlap time and the small latitude coverage, these results have limited value. The climatological latitude cross sections include the entire data sets and thus allow a more conclusive picture. In these comparisons (Fig. 5), the MIPAS and ACE-FTS δD data sets agree best between 100

and 10 hPa. Even though the biases are low in this region, the de-biased standard deviations roughly amount up to 50 per mille (larger in JJA), indicating that there are biases in the latitudinal distribution between the data sets. Also, smaller biases exist in HDO (roughly around 0.1 ppbv) and $H_2O$ (up to about 0.5 ppmv, larger in DJF), in particular in the lower part of this altitude range. They partly cancel out each other in the calculation of the δD ratio

and in the average over all latitudes, and therefore result in a relatively good agreement in δD. Above 10 hPa, the δD biases between the MIPAS and ACE-FTS data sets increase, in particular in the comparisons to the ACE-FTS v2.2 data set. Here, the MIPAS data sets show a higher depletion. The de-biased standard deviations are, however, relatively low (largely within 20 per mille) and the correlation coefficients are relatively high, indicating only small

variations over all latitudes. These δD biases are primarily driven by high biases in the MIPAS $H_2O$ data (up to 0.8 ppmv) peaking at about 2 hPa. Below 100 hPa, the MIPAS and ACE-FTS data sets also show larger deviations. The bias estimates from the seasonal (Fig. 5) and profile-to-profile comparisons differ significantly. In the profile-to-profile comparisons, the MIPAS δD data indicate a 100 per mille (20 %) higher depletion than the ACE-FTS δD

close to 200 hPa; in the climatological comparisons, on the other hand, the result is the opposite and the MIPAS data sets show significantly less depletion at 200 hPa. In both comparisons the biases are driven by differences in HDO and $H_2O$. In the profile-to-profile comparisons, the HDO biases are about 0.3–0.4 ppbv (60 % – 90 %), for $H_2O$ at they are within ±25 % at this altitude. For the climatological comparisons the HDO bias at 200 hPa is

in the range of 0.25–1 ppbv depending on season, largest bias in MAM and least in SON and DJF.



The reasons for the biases between the MIPAS and ACE-FTS data sets are manifold: differences in temporal and spatial sampling, cloud influence, vertical resolution, the choice of microwindows and spectroscopic databases as well as a less-than perfect MIPAS $H_2O$ product used here. This product is retrieved jointly with HDO in the attempt to match the

vertical resolution of the latter (see Sect. 2.2). The differences in the temporal and spatial sampling between the MIPAS and ACE-FTS observations can affect the results of climatological comparisons. This concerns, on one hand, the different measurement periods with the MIPAS observations providing data from July 2002 until March 2004 and the ACE-FTS observations starting in February 2004 and lasting for many years. In this context,

changes in variability, like trends, QBO influence, and the occurrence of sudden stratospheric warmings between these different time periods play a role. For example, according to the SWOOSH (Stratospheric Water and Ozone Satellite Homogenized, Davis et al., 2016) database, $H_2O$ has increased from the MIPAS to the ACE-FTS observation period at almost all latitudes and stratospheric altitudes. This does not help the discussion of biases among the

data sets, as the MIPAS data exhibit higher $H_2O$ mixing ratios in most of the stratosphere, in particular around 50 and 2 hPa (Fig. 7). On the other hand, the sampling issue concerns the actual sampling within a given time and latitude bin. The MIPAS satellite utilises a sun-synchronous orbit that provides more or less homogenous coverage in latitude and time. In contrast, the ACE-FTS orbit is optimised for high and middle latitudes, yielding the bulk of

observations in these regions. In this regard, the MIPAS observations provide a rather complete coverage of the individual seasons in the tropics. For the ACE-FTS orbit, the seasonal means for MAM are based on observations in April, JJA on observations in August, SON on observations in October, and DJF on observations in February, respectively. This may explain some of the tropical biases observed in Figs. 2 and 3 and is more appropriately

handled in the monthly comparisons in Sect. 4.3. Besides the actual sampling, the variability within a given time and latitude bin plays a decisive role for the sampling bias (Toohey et. al, 2013). One prominent example for this is the wintertime lower stratosphere in the Antarctic that is influenced by dehydration induced by PSCs. Such dehydration indications are visible in all data sets in both HDO and $H_2O$, but with various strengths. It is smallest in the MIPAS

data and largest in the ACE-FTS (and SMR) data sets. This behaviour can be explained by sampling biases in combination with large variability. While the MIPAS data cover all winter months in the Antarctic, the ACE-FTS coverage depends on the actual month. In June and July, the ACE-FTS observations reach only up to 60°S and 70°S, respectively. It is only in August that the observations cover latitudes polewards of 70°S, where the influence of PSCs





and the associated dehydration is substantial, leading to a pronounced signal in the seasonal average for the ACE-FTS data sets. The other explanation for the different dehydration relates to differences in cloud influence on the MIPAS and ACE-FTS observations, which actually is another aspect of a sampling bias. While the observations of the two instruments are

performed in the infrared, they use different measurement techniques. The MIPAS instrument measured thermal emission at the atmospheric limb. The solar occultation technique used by the ACE-FTS instrument provides a stronger signal, which results in a lower sensitivity to cloud presence compared to the MIPAS observations (in comparison, the SMR observations exhibit the smallest cloud influence, as they are performed in the microwave region). In the

presence of PSCs, many MIPAS observations actually have to be screened (see Introduction), weakening the dehydration signal. The other region where these differences in the cloud influence between the MIPAS and ACE-FTS data sets are important is in the troposphere. Again, more MIPAS observations are screened, which can be expected to result in a dry bias. This is observed in $H_2O$ (Fig. 7), but not in HDO (Fig. 6). In addition to this basic cloud

influence, the behaviour of HDO and $H_2O$ differs in the presence of clouds due to differences in the vapour pressure. Consequently, deviations in the depletion can be assumed, given the differences in the cloud influence between the MIPAS and ACE-FTS data sets. For example, if there are no clouds present, the dehydration near the tropical tropopause should more reflect Rayleigh fractionation and result in isotopic ratios close to -900 per mille. In the presence of

clouds, the depletion should be lower. Given that the MIPAS data sets represent more cloud-free conditions, a higher depletion relative to the ACE-FTS data sets is expected but not apparent (Fig. 8). This may relate to differences in the vertical resolution among the data sets. As described in Sect. 3.1.2, the differences in the vertical resolution of the MIPAS and ACE-FTS data sets can cause biases if they are not considered like in climatological comparisons.

Around the hygropause, the vertical resolution of the MIPAS data is roughly between 5 and 6 km, while that of the ACE-FTS data sets is between about 2 and 4 km. This contributes to the worse comparison results around the hygropause and the upper troposphere compared to core region of the stratosphere. It offers also a potential explanation why the profile-to-profile comparisons yield a better agreement, in particular in the upper troposphere where the

differences in the vertical resolution among the data sets were considered, even though the overlap is limited. The $H_2O$ bias between the MIPAS and ACE-FTS data sets, peaking at 2 hPa in the climatological comparisons, is also influenced by differences in the vertical resolution among these data sets, in particular in the upper part. This part is close to the altitude where the middle atmospheric maximum in the water vapour is typically found



(roughly 0.8 hPa or 50 km). The influence is roughly of the order of 0.1 ppmv. However, since the MIPAS data sets are worse resolved for the MIPAS $H_2O$ product used in these comparisons than the ACE-FTS data sets, they should exhibit a low bias opposite to what is observed. Hence, it can be expected that the actual biases are even larger. In addition, there

are differences in the vertical resolution of the MIPAS HDO and $H_2O$ data sets. These affect the resulting δD values and consequently cause biases, varying from tens of per mille around the hygropause to a negligible impact in the middle and upper stratosphere.

Even though the MIPAS and ACE-FTS observations are both performed in the infrared region, there are differences in the microwindows that are used to retrieve the HDO and $H_2O$

information as described in Sects. 2.2 and 2.3. These differences are larger for $H_2O$ than for HDO. The MIPAS retrievals for this special $H_2O$ product use the same microwindows as for HDO, i.e. between 6.7 and 8.0 μm, while the ACE-FTS $H_2O$ retrievals employ information from a much wider spectral region. In addition, there is also a small difference in the spectroscopic databases used in the retrieval. The MIPAS retrievals employ a special version

of the HITRAN-2000 spectroscopic database, while the ACE-FTS retrievals use HITRAN-2004 with some updates. For HDO, this difference in the spectroscopic database explains a high bias of up to 0.01 ppbv for the MIPAS data sets (Lossow et al., 2011). For $H_2O$ it may be a potential explanation for the high biases around 50 and 2 hPa observed in the MIPAS data sets. They occur during all seasons, ruling out any sampling bias origin.

Beyond that, a characteristic of the MIPAS retrieval is responsible for the obvious problems of these data (in particular for v20) in the subtropics shown in Fig. 2. This is due to the handling of the a priori data in the retrieval, which here switches as function of latitude.

The δD comparisons of data sets from the same instrument typically show smaller deviations than those of data sets from other instruments. The largest biases are observed again below

100 hPa and stem from both HDO and $H_2O$. Between the MIPAS data sets, more pronounced differences are also visible in the lower mesosphere. These biases are due to deviations in $H_2O$. As described in Sect. 2, the MIPAS data sets differ only in their calibration, while the retrieval itself was not changed. In contrast, the differences between the ACE-FTS data sets are due to changes in the instrument characterisation, choice of microwindows, and the

temperature–pressure retrieval, as detailed by Waymark et al. (2013).





### 5.3 Comparisons in the tropical lower stratosphere

Finally, the tropical comparisons exhibit in all data sets structures characteristic of a tape recorder in $\delta D$, HDO, and $H_2O$. However, there are clear quantitative differences in the isotopic/mixing ratios as well as the altitudes and the number of the structures. Overall, the

consistency in HDO and $H_2O$ is better than in $\delta D$. In February, the MIPAS data sets show a wide hygropause in HDO and $H_2O$. This is a consequence of the low temperatures in the TTL during this period, resulting in low emissions and a reduced vertical resolution of the retrieved data relative to the other months. Yet in $\delta D$, the first maximum appears well resolved. This might be a result of the different vertical resolutions of the MIPAS HDO and $H_2O$ data, or

simply a favourable interference of HDO and $H_2O$ characteristics. Differences in the vertical resolution between the MIPAS and ACE-FTS data certainly contribute to the altitude differences of tape recorder structures. However, those are also visible among the ACE-FTS data sets that have the same resolution but differ in retrieval details and the time period considered. In addition, the tape recorder structures appear to be more pronounced in the

SMR and MIPAS data sets compared to the ACE-FTS data sets. This relates to the differences in the $\delta D$ tape recorder amplitude between MIPAS and ACE-FTS data as derived from the previous studies by Steinwagner et al. (2010) and Randel et al. (2012).

## 6   Conclusion

The quality of stratospheric satellite observations of $\delta D(H_2O)$, considering five data sets, has been evaluated using profile-to-profile and climatological comparisons. The comparisons show pronounced differences in the isotopic ratio between the data sets, especially relative to the SMR data set. In the lower stratosphere, the SMR data set shows higher depletion compared to both the MIPAS and ACE-FTS data sets, with a maximum $\delta D$ bias exceeding

200 per mille close to 50 hPa. With increasing altitude, the biases get smaller in size. In the upper stratosphere and lower mesosphere, the sign of the biases between the SMR and MIPAS data sets is actually reversed. Here, the SMR data set shows less depletion, exceeding 100 per mille around 1 hPa. The $\delta D$ biases of the SMR data set are mainly driven by biases in HDO in the lower stratosphere and by biases in $H_2O$ in the upper stratosphere and lower

mesosphere. In the middle stratosphere, the $\delta D$ biases are a combination of deviations in both





HDO and H$_2$O. The dominant issues of the SMR data set are related to calibration and spectroscopic parameters.

The MIPAS and ACE-FTS data sets agree rather well, with exceptions below 100 hPa and above 10 hPa (when approaching the upper limit for the ACE-FTS observations). Between

100 and 15 hPa, the MIPAS data sets typically exhibit up to 30 per mille (10 %) less depletion, which is related to differences in both HDO and H$_2$O. Above 10 hPa, the δD biases increase and the MIPAS data sets instead show higher depletion, up to 80 per mille. These increasing biases are due to larger deviations in H$_2$O above 10 hPa. Below 100 hPa, the δD biases between the data sets are substantially larger than in the core part of the stratosphere.

Deviations between the MIPAS and ACE-FTS data sets arise from differences in the temporal and spatial sampling, cloud sensitivity, vertical resolution, and the microwindows and spectroscopic databases chosen. Different data sets from the same instrument generally agree well in the stratosphere.

*Data availability.*

The MIPAS data are available on the following website:
https://www.imk-asf.kit.edu/english/308.php
The SMR data can be accessed on the following website:
http://amazonite.rss.chalmers.se:8280/OdinSMR/searchl2

The ACE-TFS data can be downloaded from the following website:
https://databace.scisat.ca

*Competing interests.* The authors declare that they have no conflict of interests

*Acknowledgements.* This work is funded by the SNSB project Dnr 88/11 "Atmospheric modelling using space-based observations of stable water isotopes". Odin is a Swedish-led satellite project funded jointly by the Swedish National Space Board (SNSB), the Canadian Space Agency (CSA), the National Technology Agency of Finland (Tekes), and the Centre National d'Etudes Spatiales (CNES) in France. The Swedish Space Corporation has been the

prime industrial constructor. Since April 2007, Odin has been a third-party mission of ESA. We would like to thank the European Space Agency (ESA) for making the MIPAS level-1b



data set available. The Atmospheric Chemistry Experiment (ACE), also known as SCISAT, is a Canadian-led mission mainly supported by the Canadian Space Agency (CSA) and the Natural Sciences and Engineering Research Council of Canada (NSERC). S. Lossow was funded by the DFG Research Unit "Stratospheric Change and its Role for Climate Prediction"

5   (SHARP) under contract STI 210/9-2 as well as the International Meteorological Institute (IMI) hosted by the Department of Meteorology at Stockholm University. We want to express our gratitude to SPARC and WCRP (World Climate Research Programme) for their guidance, sponsorship, and support of the WAVAS-II programme.




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



| Variable | Interval | Altitude range |
|:---:|:---:|:---:|
| $\delta$D | -10000‰ – 10000‰ <br> -1250‰ – 250‰ <br> -5000‰ – 5000‰ | 1000 hPa – 70 hPa <br> 70 hPa – 1 hPa <br> 1 hPa – 0.1 hPa |
| HDO | -5 ppmv – 15 ppmv | 70 hPa – 0.1 hPa |
| $H_2O$ | -20 ppmv – 50 ppmv | 70 hPa – 0.1 hPa |

**Table 1.** Interval screening performed prior to the comparisons. Profiles that exhibited data points outside these intervals were discarded.





| Comparison | Overlap period | Latitude range in deg | Number of coincidences | Average distance separation in km | Average time separation in h | Average latitude separation in deg | Average equivalent latitude separation in deg | Average longitude separation in deg |
|---|---|---|---|---|---|---|---|---|
| SMR v2.1 vs. MIPAS v5 | Jul 2002 – Mar 2004 | 87°S – 89°N | 11564 / 13107 | 296 / 296 | 12.1 / 12.1 | 1.3 / 1.3 | 1.9 / 1.9 | 3.5 / 3.5 |
| SMR v2.1 vs. MIPAS v20 | Jul 2002 – Mar 2004 | 87°S – 87°N | 11840 / 13365 | 294 / 293 | 12.2 / 12.1 | 1.3 / 1.3 | 1.9 / 1.9 | 3.5 / 3.5 |
| SMR v2.1 vs. ACE-FTS v2.2 | Feb 2004 – May 2009 | 85°S – 86°N | 2934 / 3146 | 489 / 492 | 11.8 / 11.8 | 2.3 / 2.3 | 2.1 / 2.2 | 9 / 8.8 |
| SMR v2.1 vs. ACE-FTS v3.5 | Feb 2004 – May 2009 | 85°S – 86°N | 2963 / 3187 | 488 / 496 | 12 / 11.8 | 2.2 / 2.3 | 2.1 / 2.2 | 9.3 / 9.2 |
| MIPAS v5 vs. MIPAS v20 | Jul 2002 – Mar 2004 | 87°S – 89°N | 451619 | 0 | 0 | 0 | 0 | 0 |
| MIPAS v5 vs. ACE-FTS v2.2 | Feb 2004 – Mar 2004 | 51°N – 83°N | 384 | 552 | 17.5 | 2.2 | 2.2 | 16.5 |
| MIPAS v5 vs. ACE-FTS v3.5 | Feb 2004 – Mar 2004 | 51°N – 83°N | 364 | 568 | 17.7 | 2.2 | 2.3 | 17.3 |
| MIPAS v20 vs. ACE-FTS v2.2 | Feb 2004 – Mar 2004 | 51°N – 83°N | 373 | 577 | 17.7 | 2.2 | 2.3 | 18.1 |
| MIPAS v20 vs. ACE-FTS v3.5 | Feb 2004 – Mar 2004 | 51°N – 83°N | 352 | 585 | 17.8 | 2.1 | 2.3 | 18.6 |
| ACE-FTS v2.2 vs. ACE-FTS v3.5 | Feb 2004 – Sep 2010 | 85°S – 87°N | 19477 | 0 | 0 | 0 | 0 | 0 |

**Table 2.** Overview of the profile-to-profile comparisons and their coincidence characteristics. For comparisons to the SMR data set the characteristics vary slightly depending upon if the comparison considers HDO (upper row) or $H_2O$ (lower row). The determination of the overlap period and latitude range uses the data from both data sets.





**Figure 1.** Profile-to-profile comparisons between all data sets for for $\delta$D (top row), HDO (middle row) and $H_2O$ (bottom row). The left panels show the biases in absolute terms, while in the right panel the relative biases are shown. Please note that for the relative bias in $\delta$D the x-axis has been reversed for visual consistency with the absolute bias.





**Figure 2.** Latitude cross sections at 100 hPa from the different data sets for HDO (left column), $H_2O$ (centre column) and and $\delta$D (right column) for all seasons (different rows). The thin dashed line show the standard error of the binned data.



**Figure 3.** As Figure 2 but considering the 10 hPa level.





**Figure 4.** As Figures 2 and 3 but here for the 1 hPa level.





**Figure 5.** Summary of the seasonal comparisons that were shown exemplarily in Fig. 2 to 4. Here δD is considered. In the left panels the correlations between the latitudinal cross sections are shown for the individual seasons. The middle panels show the bias averaged over all latitudes and right panels the corresponding de-biased standard deviation.





**Figure 6.** As Fig. 6 but here for HDO.




**Figure 7.** As Fig. 6 but here for H$_2$O.



**Figure 8.** Tropical mean (15°S – 15°N) profiles for δD (left column), HDO (middle column) and H$_2$O (right column) for February, April, August and October given in different rows.