# Peer review of "The SPARC water vapour assessment II: Profile-to-profile and climatological comparisons of stratospheric $\delta D(H_2O)$ observations from satellite"

_Atmospheric Chemistry and Physics, 2018_

## Referee Comment (RC1) · D. G. Johnson (Referee) · 19 Jul 2018

Summary:

1) Does the paper address relevant scientific questions within the scope of ACP? Yes.
2) Does the paper present novel concepts, ideas, tools, or data? Yes (new comparisons of available data sets). 3) Are substantial conclusions reached? Qualitative; more quantitative conclusions desired, see explanation and more detailed comments below. 4) Are the scientific methods and assumptions valid and clearly outlined? Yes,

for the most part, see detailed comments below. 5) Are the results sufficient to support the interpretations and conclusions? See response to 3). 6) Is the description of experiments and calculations sufficiently complete and precise to allow their reproduction by fellow scientists (traceability of results)? Yes. 7) Do the authors give proper credit to related work and clearly indicate their own new/original contribution? Yes. 8) Does the title clearly reflect the contents of the paper? Yes. 9) Does the abstract provide a concise and complete summary? Yes. 10) Is the overall presentation well structured and clear? Yes. 11) Is the language fluent and precise? Yes. 12) Are mathematical formulae, symbols, abbreviations, and units correctly defined and used? Yes. 13) Should any parts of the paper (text, formulae, figures, tables) be clarified, reduced, combined, or eliminated? See specific comments below for minor clarification requests. 14) Are the number and quality of references appropriate? Yes. 15) Is the amount and quality of supplementary material appropriate? Yes.

General comments and overall impression: The authors have summarized a large body of work in comparing profiles of HDO, H2O, and the ratio made by 3 instruments (SMR, MIPAS, and ACE-FTS), with two retrieval versions each for MIPAS and ACE-FTS. I feel strongly that this work is worth publishing, but in my opinion (and others may reasonably disagree) it is also incomplete. While I provided many comments below, my intent is not to be overly critical, but to suggest additional content so as to increase the return from the effort that the authors have already put into this manuscript.

I believe that the primary reason that one compares measurements made by multiple instruments is to validate a priori estimates of measurement accuracy. This validation is imperfect, and the comparison cannot be used to estimate accuracy since the difference between measured values does not tell you if any one value is correct. However, although I may have missed it, I did not see a discussion of the retrieval accuracy for any of the three instruments, only precision, so I do not know if the differences that were discussed were consistent with estimates of individual biases. And in the end, there were so many reasons for differences in the profiles (differences in spectral databases,

calibration errors, time/space coverage, and so on), that that I am not sure that I have learned anything about the absolute accuracy of the various measurements of H2O, HDO, and delta D.

I would ask the authors to: 1) Provide a quantitative discussion of sources of systematic error for HDO, H2O, and delta D (some errors that affect individual profiles will cancel in the ratio, others will not) for each instrument. Perhaps provide an error budget in a table for each molecule and instrument. Or, if the paper is already struggling with length, maybe put it into the supplement and at least include a rolled up estimate that can be compared to the observed differences between instruments. 2) Try to say something conclusive about what was learned from the comparisons about the quality of the data. Plausible qualitative explanations are provided for differences observed in various regions (at the end of the Conclusion, for example), but I still don't know which profile to believe.

An additional personal preference: I feel that the comparisons are further complicated by showing two retrieval versions each for MIPAS and ACE-FTS. I would prefer sticking with the latest release (not beta or test) version. A simple comparison between versions for each instrument might be called for if there is an extensive publication record for the older version and it is necessary to show the difference, but it would simplify things to show only 3 comparisons (3 pairs selected from a set of 3 data sets) instead of 10 (10 pairs selected from a set of 5 data sets).

Specific comments: page 11, line 5, and page 34, line 15: Although this information is probably in the references (and citations in those references), it would be useful to include some discussion of the specific differences in the spectral databases used for MIPAS and ACE-FTS (and SMR, for that matter). Specifically, whose line parameters (strengths, positions, and linewidths) are used for H2O and HDO? What are the uncertainties in the parameters for the lines used in the retrievals, and how does that affect the profiles?

page 16, line 9-10: This would probably be obvious to most people, but it would have been helpful to me to clarify to me here that by "all available data" you meant (I assume) the full data sets for each instrument as described in section 2. As distinct from the subsets used for profile comparisons as listed in table 2.

page 30, lines 13-31: This was a section that I thought needed to be more quantitative. Sideband leakage is specified, but the bias this may cause in $H_2O$ is not quantified. Likewise, bias due to spectroscopic parameters is mentioned, but the parameters are not identified, the uncertainty in the parameters is not specified, and the effect on retrievals is not quantified. Having this additional information is very useful when trying to make sense of the difference between SMR and other sensors.

page 32, line 18: when the authors specified "homogeneous coverage in latitude and time", I was confused. A sun-sync orbit covers all latitudes but just 2 times (ascending and descending) at each latitude. Does time mean season, not time of day?

Figure 1, lower left panel ($H_2O$ bias): This figure confused me. Looking at around 30 hPa, we get SMR-MIPAS~-1.4 ppmV, and SMR-ACE~-0.7 ppmV. That would suggest that MIPAS-ACE~+0.7ppmV. But the figure shows more like +/-0.2 ppmV (depending on the exact algorithm pair). However, Figure 7 shows MIPAS-ACE much closer to +0.7 ppmV, even though this is not a direct profile-profile comparison. Does this suggest that, for $H_2O$ anyway, the direct comparison of ACE and MIPAS is invalid due to insufficient data and poor coincidence?

Grammatical: page 6 of 55, line 9: should this read "climatological comparisons"?
* * *

---

## Referee Comment (RC2) · Anonymous Referee #2 · 9 Aug 2018

Review of "The SPARC water vapour assessment II: Profile-to-profile and climatological comparisons of stratospheric $\delta$D(H2O) observations from satellite" by Charlotta Högberg et al.

Högberg et al. compare $\delta$D(H$_2$O) observations from three different satellites in the stratosphere. The paper is well written and it provides a detailed analysis of H$_2$O, HDO and $\delta$D(H$_2$O) measurements from the three instruments. Such work is important for the scientific use of these kind of data sets and I really think the paper should be published, however, my reason for clicking on "reject" mainly bases on the authors' journal choice.

[Figure]

I.e. to my understanding, this is not an "ACP paper", but see my main point below for more details on this. A few minor points that should be considered before publication are also listed below.

**Major point**

- In my opinion, an ACP paper is supposed to tackle a question/issue/problem of a physical or a chemical process in the atmosphere. That is not done at all in this study. It is (merely) the comparison of three different data products. A lot can be learned from this about retrieval parameters, about effects of spatial and temporal sampling or of using different frequency bands, and so on, but here nothing is learned about the physics or the chemistry of the atmosphere. Hence, I suggest to withdraw the paper from ACP and to submit it e.g. to AMT or a related journal. Those journals are also well-renowned and provide room for rather technical papers like this one here.

**Minor points**

- P3L31andL32: "boreal winter and "boreal" summer

- P4L7-8: That is very simplified, the issue is a lot more nuanced. Please see Frank et al. 2018, 10.5194/acp-18-9955-2018

- P5L10: There should be more recent literature for this than from 1996

- P5L28 and P6L5: Eichinger et al. 2015 (10.5194/acp-15-5537-2015) dealt with this issue in a model-satellite comparison

- P6L16: Explain what LT means.

- Sect. 4: Why do you start with showing biases and not the actual profiles first? I find that confusing.

- I agree with what Mr. Johnson says that it is hard to say what can actually be learned from this study, since there are so many differences between the different methods, one cannot actually see any causalities. I would also appreciate some sort of conclusion that at least states this product/instrument is better here, and this is worse there. Which product can be trusted more where and/or when for making process studies or model comparisons? And maybe also methodologically, which method is best for what? In a future satellite mission, how would the "best" instrument look like, and how can the retrievals be improved?

- For several reasons the paper is pretty lengthy and that could easily be reduced: The information in Sects. 2 and especially 3 should be reduced to the most important points, technical details and the bulk of the equations should be banished to the supplement. Moreover, the paper is pretty repetitive, e.g. the (first part of the) discussion and the conclusion are not more than summaries of the results. Some restructuring and removing can easily resolve this.

---

## Author Comment (AC1) · 13 Nov 2018

The authors thank David G. Johnson for constructive comments and suggestions for revision. In the following our replies are given in blue, we have revised the manuscript accordingly.

Replies to review #1 comments:

General comments and overall impression:

The authors have summarized a large body of work in comparing profiles of HDO, $H_2O$, and the ratio made by 3 instruments (SMR, MIPAS, and ACE-FTS), with two retrieval versions each for MIPAS and ACE-FTS. I feel strongly that this work is worth publishing, but in my opinion (and others may reasonably disagree) it is also incomplete. While I provided many comments below, my intent is not to be overly critical, but to suggest additional content so as to increase the return from the effort that the authors have already put into this manuscript.

General comment #1: I believe that the primary reason that one compares measurements made by multiple instruments is to validate a priori estimates of measurement accuracy. This validation is imperfect, and the comparison cannot be used to estimate accuracy since the difference between measured values does not tell you if any one value is correct. However, although I may have missed it, I did not see a discussion of the retrieval accuracy for any of the three instruments, only precision, so I do not know if the differences that were discussed were consistent with estimates of individual biases. And in the end, there were so many reasons for differences in the profiles (differences in spectral databases, calibration errors, time/space coverage, and so on), that I am not sure that I have learned anything about the absolute accuracy of the various measurements of $H_2O$, HDO, and delta D.

General response #1: Our work is part of a larger assessment. These comparisons aim to provide a contemporary picture of the typical differences in the observational database of δD and to draw some conclusions what these differences may mean for scientific analysis. It is not a validation of a single data set, aiming to validate its a priori estimates of accuracy. This aim is now much more clearly stated. Arguably, the database for δD is sparse, relative to that of $H_2O$. This leaves some room to address issues of individual data sets. We tried this to the best of the data we had available. Clearly multiple aspects play a role for the differences among the data sets, there is no simple attribution to a single cause.

I would ask the authors to:

General comment #2: Provide a quantitative discussion of sources of systematic error for HDO, $H_2O$, and delta D (some errors that affect individual profiles will cancel in the ratio, others will not) for each instrument. Perhaps provide an error budget in a table for each molecule and instrument. Or, if the paper is already struggling with length, maybe put it into

the supplement and at least include a rolled up estimate that can be compared to the observed differences between instruments.

General response #2: There are several sources for the systematic errors for each instrument and it is difficult to state the exact contribution for each of them. It is complicated to get the full picture, but we have tried to include the known issues effecting the retrievals e.g. spectral database, calibration or sampling error (for the zonal and seasonal means) where applicable. In the revised version we expanded on that based on new retrieval tests for MIPAS and SMR.

A $\delta D$ error budget as such only exists for MIPAS. It is a common problem for many comparisons that such information do not exist for all data sets. For the MIPAS v5 data set the total error of stratospheric $\delta D$ has been estimated to be about 100 - 150 per mille (Steinwagner at al., 2007). However, this is based on one retrieval in the tropics only. It provides a measure of accuracy, but it may not be applicable for latitudes outside the tropics. Nevertheless, this information has been added to the data set description.

General comment #3: Try to say something conclusive about what was learned from the comparisons about the quality of the data. Plausible qualitative explanations are provided for differences observed in various regions (at the end of the Conclusion, for example), but I still don't know which profile to believe.

General response #3: As stated above the aim of this study was to provide a picture of the typical differences in the observational database of $\delta D$.

All data sets have their strengths and weaknesses. Which data set is the most reliable can depend on a number of aspects, like application, time, altitude or latitude of interest, for example. From the data we have, we would generally rate the ACE-FTS data sets as the best. For them the least number of systematic errors have been identified, yet that does not mean that they are perfect. Version 3.5 is an improvement over version 2.2 and has the benefit of a larger altitude range and time period covered. Behind that we would rate the MIPAS data sets. Here primarily the differences in the vertical resolution of the HDO and $H_2O$ data used for the calculation of $\delta D$ cause concern. A rating between the two MIPAS data sets is difficult, there is nothing much between them. The largest issues occur for the SMR data set, relating primarily to the characterisation of the sideband filtering and the issues in the spectroscopy. Despite that, this data set shows good consistency in latitudinal cross sections, for example.

General comment #4: An additional personal preference: I feel that the comparisons are further complicated by showing two retrieval versions each for MIPAS and ACE-FTS. I would prefer sticking with the latest release (not beta or test) version. A simple comparison between versions for each instrument might be called for if there is an extensive publication record for the older version and it is necessary to show the difference, but it would simplify things to show only 3 comparisons (3 pairs selected from a set of 3 data sets) instead of 10 (10 pairs selected from a set of 5 data sets).

General response #4: None of the data sets are beta or test versions. The ACE-FTS v2.2 and MIPAS v5 data sets has been used in previous studies (e.g. Randel et al., 2012; Lossow et al., 2011). The ACE-FTS v3.5 data set covers a longer observational period compared to v2.2. Also, the microwindows have been optimised allowing HDO and $H_2O$ information to be retrieved at higher altitudes, as described in Sect. 2. The MIPAS v20 data set is based on an improved calibration of the spectra provided by the European Space Agency (ESA).

Within the WAVAS assessment we had an open data policy. Everyone was invited, to get a coverage of the observational database from satellites as complete as possible. The different instrument teams decided which data set versions that should be included in the evaluations. In accordance to that, the decision was made to include two MIPAS and ACE-FTS data sets to provide an overview of the differences between the well-validated older versions compared to the newer versions.

Specific comments:

Specific comment #1: page 11, line 5, and page 34, line 15: Although this information is probably in the references (and citations in those references), it would be useful to include some discussion of the specific differences in the spectral databases used for MIPAS and ACE-FTS (and SMR, for that matter). Specifically, whose line parameters (strengths, positions, and linewidths) are used for $H_2O$ and HDO? What are the uncertainties in the parameters for the lines used in the retrievals, and how does that affect the profiles?

Specific response #1: Since, both, ACE-FTS and MIPAS use multiple microwindows we feel that listing all lines with its parameters is a little bit too much. For ACE-FTS v2.2 the information on the considered microwindows is found in the following document: https://database.scisat.ca/level2/ace_v2.2/ACE-SOC-0020-microwindow_list_for_v2.2_and_updates_Dec1.pdf. The corresponding document for v3.5 is located here: https://database.scisat.ca/level2/ace_v3.5_v3.6/ACE-SOC-0027-ACE-FTS_Spectroscopy-version_3.5_Jan222016_Rev1A.pdf. Please note that a registration is required to obtain these documents. For MIPAS the microwindows are listed in Tab. 1 of Steinwagner et al. (2007).

In the revised version we have quantified the effect of the different spectral databases used in the ACE-FTS and MIPAS retrievals. The uncertainty of the different spectroscopic parameters is typically a few percent. We have performed tests with the SMR retrieval to quantify the effect of these uncertainties, assuming a 5% uncertainty in the line broadening parameter, a 10% uncertainty temperature dependence exponent and a 2% uncertainty line intensity.

Specific comment #2: page 16, line 9-10: This would probably be obvious to most people, but it would have been helpful to me to clarify to me here that by "all available data" you meant (I assume) the full data sets for each instrument as described in section 2. As distinct from the subsets used for profile comparisons as listed in table 2.

Specific response #2: With "all available data" is meant that the complete data sets are considered. The text has been changed to make that clearer.

Specific comment #3: page 30, lines 13-31: This was a section that I thought needed to be more quantitative. Sideband leakage is specified, but the bias this may cause in $H_2O$ is not quantified. Likewise, bias due to spectroscopic parameters is mentioned, but the parameters are not identified, the uncertainty in the parameters is not specified, and the effect on retrievals is not quantified. Having this additional information is very useful when trying to make sense of the difference between SMR and other sensors.

Specific response #3: We have expanded this section in the revised version. Besides the retrieval tests regarding the uncertainty of spectroscopic parameter in the SMR retrieval, as mentioned in the specific response #1, we have also performed retrieval test focusing on the sideband leakage.

Specific comment #4: page 32, line 18: when the authors specified "homogeneous coverage in latitude and time", I was confused. A sun-sync orbit covers all latitudes but just 2 times (ascending and descending) at each latitude. Does time mean season, not time of day?

Specific response #4: The sampling differences were discussed in the context of seasonal comparisons. In that sense time means season. The MIPAS observations cover almost all latitudes twice a day. Over the course of a season this coverage is rather homogenous (the longitudes vary). In contrast, the ACE-FTS observations focus on high and middle latitudes and therefore have most of the observations in that region. On a given day only two latitudes are covered (typically one in the Northern Hemisphere, one in the Southern Hemisphere). During a season the covered latitudes vary.

Specific comment #5: Figure 1, lower left panel (H2O bias): This figure confused me. Looking at around 30 hPa, we get SMR-MIPAS←-1.4 ppmV, and SMR-ACE←-0.7 ppmV.

That would suggest that MIPAS-ACE←+0.7ppmV. But the figure shows more like +/-0.2 ppmV (depending on the exact algorithm pair). However, Figure 7 shows MIPAS-ACE much closer to +0.7 ppmV, even though this is not a direct profile-profile comparison. Does this suggest that, for $H_2O$ anyway, the direct comparison of ACE and MIPAS is invalid due to insufficient data and poor coincidence?

Specific response #5: Essentially, these results cannot be combined in a commutative manner.

The profile-to-profile comparisons (Fig. 1) among the different data sets consider different time periods and latitude bands (especially the comparisons between the ACE-FTS and MIPAS data sets, see Tab. 2). Therefore, it is already in this case not possible to deduce a specific comparison result from the combination of other comparisons.

The seasonal comparisons (Fig. 2-7) consider the complete data sets and extend the results from the profile-to-profile comparisons shown in Fig.1. This is especially relevant for the comparisons between the MIPAS and ACE-FTS data sets, which have a limited overlap time.

Given, these different conditions the comparison results cannot easily be combined or transferred from one figure to another.

Specific comment #6: Page 6 of 55, line 9: should this read "climatological comparisons"?

Specific response #6: The text has been changed to "climatological comparisons".

References

Lossow, S., et al., Comparison of HDO measurements from Envisat/MIPAS with observations by Odin/SMR and SCISAT/ACE-FTS, Atmospheric Measurement Techniques, 4, 1855 – 1874, doi:10.5194/amt-4-1855-2011, 2011.

Randel, W. J., E. Moyer, M. Park, E. Jensen, P. Bernath, K. Walker, and C. Boone, Global variations of HDO and HDO/$H_2O$ ratios in the upper troposphere and lower stratosphere derived from ACE-FTS satellite measurements, Journal of Geophysical Research, 117(D16), D06,303, doi:10.1029/2011JD016632, 2012.

Steinwagner, J., M. Milz, T. von Clarmann, N. Glatthor, U. Grabowski, M. Höpfner, G. P. Stiller, and T. Röckmann, HDO measurements with MIPAS, Atmospheric Chemistry & Physics, 7, 2601 – 2615, doi:10.5194/acp-7-2601-2007, 2007.

---

## Author Comment (AC2) · 13 Nov 2018

The authors thank referee #2 for constructive comments and suggestions for revision. In the following our replies are given in blue, we have revised the manuscript accordingly.

Replies to review #2 comments:

Major point

General comment #1: In my opinion, an ACP paper is supposed to tackle a question/issue/problem of a physical or a chemical process in the atmosphere. That is not done at all in this study. It is (merely) the comparison of three different data products. A lot can be learned from this about retrieval parameters, about effects of spatial and temporal sampling or of using different frequency bands, and so on, but here nothing is learned about the physics or the chemistry of the atmosphere. Hence, I suggest to withdraw the paper from ACP and to submit it e.g. to AMT or a related journal. Those journals are also well-renowned and provide room for rather technical papers like this one here.

General response #1: In the revised version we added more information on atmospheric processes. In particular we reflect more on what the differences among the data sets mean for our ability to learn something about the atmosphere.

Minor points

Specific comment #1: P3L31andL32: "boreal winter and "boreal" summer.

Specific response #1: The text has been changed.

Specific comment #2: P4L7-8: That is very simplified, the issue is a lot more nuanced. Please see Frank et al. 2018, 10.5194/acp-18-9955-2018.

Specific response #2: We agree. The description of the variation of the amount of water vapour produced by oxidation of methane in the stratosphere has been adjusted and the work of le Texier et al. (1988) and Frank et al. (2018) have been referenced.

Specific comment #3: P5L10: There should be more recent literature for this than from 1996.

Specific response #3: Yes, there is. Moyer et al. (1996) is still used as reference since it is the seminal paper regarding satellite observations of deuterated water and their application to study the transport of water vapour from the troposphere into the stratosphere. We have now also added newer references, as Nassar et al. (2007), Payne et al. (2007), Sayres et al. (2010), Steinwagner et al. (2010) and Eichinger et al. (2015).

Specific comment #4: P5L28 and P6L5: Eichinger et al. 2015 (10.5194/acp-15-5537-2015) dealt with this issue in a model-satellite comparison.

Specific response #4: This text passage concerns δD comparisons among satellite data sets. In that sense involving the work of Eichinger et al. (2015), that focuses primarily on model simulations, is not really wanted here. Arguably this work helps to understand the observational discrepancies in the δD tape recorder to some degree. However, we have chosen not to make this aspect a major topic of this work. This will be addressed in much more detail in a different study.

Specific comment #5: P6L16: Explain what LT means.

Specific response #5: LT stands for local time. The abbreviation has now been defined in the text.

Specific comment #6: Sect. 4: Why do you start with showing biases and not the actual profiles first? I find that confusing.

Specific response #6: We agree it would be better to show some profiles before the description of the biases. Showing the actual profiles for all comparisons in a single figure is difficult. With 10 comparisons you end up with 20 profiles, making it hard to distinguish anything. Therefore, we show exemplarily the profiles for one comparison in the main manuscript. The profiles for the remaining 9 comparisons are shown in the supplement.

Specific comment #7: I agree with what Mr. Johnson says that it is hard to say what can actually be learned from this study, since there are so many differences between the different methods, one cannot actually see any causalities. I would also appreciate some sort of conclusion that at least states this product/instrument is better here, and this is worse there. Which product can be trusted more where and/or when for making process studies or model comparisons? And maybe also methodologically, which method is best for what? In a future satellite mission, how would the "best" instrument look like, and how can the retrievals be improved?

Specific response #7: As stated in our general response #3 to the first reviewer all data sets have their strengths and weaknesses and that aspects like application, time, altitude or latitude can influence which data sets is better or worse. In the current situation we rate the ACE-FTS data sets as the most reliable in terms of δD.

The biggest concern is probably not to improve the retrieval method itself. There is always the attempt to have the best retrieval with the best characterisation of the instrument. An ideal instrument should have high sensitivity, good coverage spectrally and in space and time, is little influenced by clouds and aerosols, etc. ACE-FTS has a very good signal-to-noise ratio, but a limited temporal and spatial coverage. For MIPAS and SMR coverage is a strength, but the signal-to-noise ratio is clearly smaller than for ACE-FTS. The MIPAS observations are clearly influenced by clouds and aerosols, while the SMR observations are almost insensitive

to that. In that sense no single instrument is perfect. What you define as best depends on what you want to achieve.

Specific comment #8: For several reasons the paper is pretty lengthy and that could easily be reduced: The information in Sects. 2 and especially 3 should be reduced to the most important points, technical details and the bulk of the equations should be banished to the supplement. Moreover, the paper is pretty repetitive, e.g. the (first part of the) discussion and the conclusion are not more than summaries of the results. Some restructuring and removing can easily resolve this.

Specific response #8: We prefer to keep data set (Sect. 2) and comparison approach (Sect. 3) description detailed, as done now. Within the water vapour assessment the minor isotopologue data sets are not described in detail elsewhere as done for the main isotopologue (Walker and Stiller et al., in preparation). Given the assessment role, our results should be easily reproducible by others. For that the approach description needs to a have certain level of detail and transparency. Arguably, the calculation approach for δD has become more complicated because of the necessary adjustments of the MIPAS and ACE-FTS data sets to the SMR data set, which do not measure HDO and $H_2O$ simultaneously. However, that should not be brushed over.

Most concerned we are with the conclusion section, which feels quite repetitive with respect to the discussion. We have opted to merge these two sections.

References

Eichinger, R., P. Jöckel, and S. Lossow, Simulation of the isotopic composition of strato- spheric water vapour – Part 2: Investigation of HDO / $H_2O$ variations, Atmospheric Chemistry & Physics, 15, 7003 – 7015, doi:10.5194/acp-15-7003-2015, 2015.

Frank, F., P. Jöckel, S. Gromov, and M. Dameris, Investigating the yield of $H_2O$ and $H_2$ from methane oxidation in the stratosphere, Atmospheric Chemistry & Physics, 18, 9955 – 9973, doi:10.5194/acp-18-9955-2018, 2018.

Le Texier, H., S. Solomon, and R. R. Garcia, The role of molecular hydrogen and methane oxidation in the water vapour budget of the stratosphere, Quarterly Journal of the Royal Meteorological Society, 114, 281 – 295, doi:10.1002/qj.49711448002, 1988.

Moyer, E. J., F. W. Irion, Y. L. Yung, and M. R. Gunson, ATMOS stratospheric deuter- ated water and implications for troposphere-stratosphere transport, Geophysical Research Letters, 23, 2385 – 2388, doi:10.1029/96GL01489, 1996.

Nassar, R., P. F. Bernath, C. D. Boone, A. Gettelman, S. D. McLeod, and C. P. Rinsland, Variability in HDO/$H_2O$ abundance ratios in the tropical tropopause layer, Journal of Geophysical Research, 112(D11), D21,305, doi:10.1029/2007JD008417, 2007.

Payne, V. H., D. Noone, A. Dudhia, C. Piccolo, and R. G. Grainger, Global satellite measurements of HDO and implications for understanding the transport of water vapour into the stratosphere, Quarterly Journal of the Royal Meteorological Society, 133 (627), 1459 – 1471, doi:10.1002/qj.127, 2007.

Sayres, D. S., et al., Influence of convection on the water isotopic composition of the tropical tropopause layer and tropical stratosphere, Journal of Geophysical Research, 115(D14), D00J20, doi:10.1029/2009JD013100, 2010.

Steinwagner, J., S. Fueglistaler, G. P. Stiller, T. von Clarmann, M. Kiefer, P. Borsboom, A. van Delden, and T. Röckmann, Tropical dehydration processes constrained by the seasonality of stratospheric deuterated water, Nature Geoscience, 3, 262 – 266, doi: 10.1038/ngeo822, 2010.

Walker, K. A. and Stiller, G. P.: The SPARC water vapour assessment II: Data set overview, in preparation.

---

## Editor Decision (ED1)

Note, I'm referencing line and page numbers from the version found in the author's response.

**ABSTRACT**
Change "Our focus is on stratospheric altitudes" to "Our primary focus is on stratospheric altitudes"
Change "of the isotopic ratio, which primarily concerns the comparisons of the MIPAS and ACE-FTS
data sets to the SMR data set." To "of the isotopic ratio, mainly in regards to comparisons between the SMR data and both MIPAS and ACE-FTS."
Change ", this data set" to ", the SMR data set"
On line 15, delete "typically"
On line 19: change "the biases in δD are a combination of deviations in both HDO and H2O." to "the biases in δD are the result of deviations in both HDO and H2O."
Line 24: delete "typically"
Line 29: change " The differences between the MIPAS and ACE-FTS data come from different aspects," to "The differences between the MIPAS and ACE-FTS data have multiple causes, "

This sentence is confusing "Overall, if the data sets are combined, the δD differences among them in key areas of scientific interest (e.g. tropical and polar lower stratosphere, lower mesosphere and upper troposphere) make it currently rather difficult to draw robust conclusions on atmospheric processes affecting the water vapour budget and distribution, e.g. the individual contribution from different transport mechanisms of water vapour into the stratosphere."

Rewrite instead to (assuming I've interpreted your analysis correctly) to
"The differences in δD between the various data sets are sufficiently large to be able to draw robust conclusions on atmospheric processes affecting the water vapour budget and distribution. The bottom line is we can't make definitive statements regarding possibly transport mechanisms of water vapour into the stratosphere based upon the δD date assessed here."

**INTRO**
page 14: line 8, change "to a large part" to "in a large part"

page 15: line 2/3, change "is preserved up to about 30 km before it dissipates." to " is traceable up to ~30 km in the isolated tropical pipe region, above which it mixes out."

page 15, line 19/20: change "Despite their low abundance, these minor isotopologues are important for atmospheric science, as they eventually can provide additional information in the form of isotopic ratios relative to the main
isotopologue, H216O (hereafter named H2O)." To "Although found in low abundance, the minor isotopologues can providee information on the process history of air parcels from their isotopic ratios relative to the main isotopologue ($H_2^{16}O$) (hereafter called $H_2O$)."

page 16, line 12-14: this sentence isn't clear "If the dehydration due to the slow ascent of air through the TTL is considered on its own (which corresponds to a Rayleigh fractionation process), a δD value around –900 per mille would be expected near the tropopause." I'd rewrite

as "If air dehydrates to the saturation mixing ratio as it slowly ascends through the TTL, undergoing a Rayleigh fractionation process, a δD value around −900 per mille would be expected near the tropopause."

page 16, line 20:  change "with primary" to "with a primary"

page 16: line 32/33:  change "In terms of δD no such comparisons exist among the satellite observations." To "There are not published comparisons of δD between available satellite observations."

page 17: line 8/9:  text states "This observational discrepancy remains unresolved up to now." Are you going to resolve that discrepancy in this paper?

**2: DATA SETS**
Can you add a statement akin to your answer to a reviewer that you consider multiple versions from the same satellite instrument because they have been published in the past  and this provides context for older studies as well as future studies?

2.1 Odin/SMR  page 18, line 2/3  I would change" from pole to pole. This particularly concerns the boreal winter time."  To "from pole to pole, mainly during boreal winter."

2.1 Odin/SMR  page 18, line 11. Change "with one orbit covering HDO, followed by one orbit covering the H2O" to "with one orbit measuring HDO, followed by one orbit measuring the H2O"

2.1 Odin/SMR  page 19, line 2 change "shall" to "should"  (and, can you say what the quality categories mean?)

2.1 Odin/SMR  page 19, line 17, change "drift, which are" to "drift, which is"

2.2 Envisat/MIPAS…page 20, line 21, add a comma after "(Flaud et al., 2003)"  (before the "which")

2.2 Envisat/MIPAS…page 20, line 27,. Add a comma after "Steinwagner et al. (2007)"

2.2 Envisat/MIPAS…page 20, line 32, add a comma after "For δD"

2.2 Envisat/MIPAS…page 21…line 1, this is first note of the visibility flag".   Can  you say what it is (as you did for the screening for SMR at the top of page 19)?

2.2 Envisat/MIPAS, page 21, line 9.  You can delete "However, the impact on this work is negligible."

2.3 SCISAT/ACE-FTS page 22 line 21, change "utilized" to "used"

**3: APPROACH**
page 23 line 1 / 2  change "stratosphere. As a complement, we also use data for the upper troposphere and lower mesophere." To "stratosphere, although we use data for the upper troposphere and lower mesosphere where available."

page 23 line 6-10;  rewrite as
(1) Calculate δD from the individual H2O and H2O profiles, then average.  Here we denote this approach as "individual."
(2) First calculate average H2O and H2O profiles, and then calculate δD.  Here we refer to this approach as "separate".

3.1 profile to profile comparisons page 25, line 14..change Haymsfield to Heymsfield

3.1 page 25, line 10-17….I understand that you can't use equation 3 for delta-d, but can't you use it on the HDO and H2O before calculating delta-d?

3.1.4, line 8, change "concerns mostly" to "impacts mainly"

3.3, page 29, line 17, delete "in particular"

**4: RESULTS**
4.1, page 30, This rewrite is what was needed to add more science to the comparisons.   Similarly the addition on page 33/34 was needed.

Page 35, line 9, delete "typically"

Page 36, delta-d section, first paragraph...I'm confused by the discussion here.  Some referencing to the figures for the seasonal as well as the figure 2 in the profile comparisons might help.

(also, in captions where you show biases, even though you say in the text, also say in the caption that the bias is A-B (or B-A) rather A vs B.

Page 38, line 21...instead of "more suited" just say "shown"  (you need to say "more suited to something",  not just "more suited"

**5 DISCUSSION AND CONCLUSION**
page 41, line 8/9  states " This concerns primarily the comparisons to the SMR data set." which is awkward.  I suggest "These quantitative differences are largest with comparisons to the SMR data."

page 41, line 11-14 says " In the profile-to-profile comparisons (Fig. 2), the SMR data set shows in the lower stratosphere a significantly higher depletion in δD than the MIPAS and ACE-FTS data sets, which however is close to its lower limit, thus correspondingly the uncertainties are larger."  Also, pretty awkward sentence construction.  I suggest "For the profile-to-profile comparisons (Fig. 2), in the lower stratosphere the SMR data set shows significantly higher δD

depletions than for the MIPAS and ACE-FTS data sets.  However, we note that this is close to the SMR lower limit, which has larger uncertainties."

page 46, line 17/18 says " As described in Sect. 3.1.2, the differences in the vertical resolution of the MIPAS and ACE-FTS data sets can cause biases if they are not considered like in climatological comparisons."  See my question above regarding section 3.1 " page 25, line 10-17".  It really seems that you need to consider the resolution differences for any type of comparison.  (and, if you leave this sentence alone, change "like in climatological comparison" to "as with the climatological comparisons".

page 49, line 6, change "data we have at hand the least number of systematic errors" to "data currently available, fewer systematic error"

page 49, line 10, change "principal" to "principle"

---

## Author Response (AR2)

Note, I'm referencing line and page numbers from the version found in the author's response.

Replies to editor comments:

**ABSTRACT**

Change "Our focus is on stratospheric altitudes" to "Our primary focus is on stratospheric altitudes"
Change "of the isotopic ratio, which primarily concerns the comparisons of the MIPAS and ACE-FTS data sets to the SMR data set." To "of the isotopic ratio, mainly in regards to comparisons between the SMR data and both MIPAS and ACE-FTS."
Change ", this data set" to ", the SMR data set"
On line 15, delete "typically"

On line 19: change "the biases in δD are a combination of deviations in both HDO and $H_2O$." to "the biases in δD are the result of deviations in both HDO and $H_2O$."
Line 24: delete "typically"
Line 29: change "The differences between the MIPAS and ACE-FTS data come from different aspects," to "The differences between the MIPAS and ACE-FTS data have multiple causes, "

Response: The points for the Abstract outlined above have been changed in the text according to the suggestions.

This sentence is confusing "Overall, if the data sets are combined, the δD differences among them in key areas of scientific interest (e.g. tropical and polar lower stratosphere, lower mesosphere and upper troposphere) make it currently rather difficult to draw robust conclusions on atmospheric processes affecting the water vapour budget and distribution, e.g. the individual contribution from different transport mechanisms of water vapour into the stratosphere."

Rewrite instead to (assuming I've interpreted your analysis correctly) to
"The differences in δD between the various data sets are sufficiently large to be able to draw robust conclusions on atmospheric processes affecting the water vapour budget and distribution. The bottom line is we can't make definitive statements regarding possibly transport mechanisms of water vapour into the stratosphere based upon the δD date assessed here."

Response: The text has been rewritten to "Overall, if the data sets are considered together, the differences in δD among them in key areas of scientific interest (e.g. tropical and polar lower stratosphere, lower mesosphere and upper troposphere) are too large to draw robust conclusions on atmospheric processes affecting the water vapour budget and distribution, e.g. the relative importance of different mechanisms transporting water vapour into the stratosphere. "

**INTRO**

page 14: line 8, change "to a large part" to "in a large part"

page 15: line 2/3, change "is preserved up to about 30 km before it dissipates." to " is traceable up to ~30 km in the isolated tropical pipe region, above which it mixes out."

page 15, line 19/20: change "Despite their low abundance, these minor isotopologues are important for atmospheric science, as they eventually can provide additional information in the form of isotopic ratios relative to the main isotopologue, $H_2^{16}O$ (hereafter named $H_2O$)." To "Although found in low abundance, the minor isotopologues can provide information on

the process history of air parcels from their isotopic ratios relative to the main isotopologue $(H_2^{16}O)$ (hereafter called $H_2O$)."

page 16, line 12-14: this sentence isn't clear "If the dehydration due to the slow ascent of air through the TTL is considered on its own (which corresponds to a Rayleigh fractionation process), a $\delta D$ value around –900 per mille would be expected near the tropopause." I'd rewrite as "If air dehydrates to the saturation mixing ratio as it slowly ascends through the TTL, undergoing a Rayleigh fractionation process, a $\delta D$ value around –900 per mille would be expected near the tropopause."

page 16, line 20: change "with primary" to "with a primary"

Response: The outlined points above for the introduction have been changed in the text according to the suggestions.

page 16: line 32/33: change "In terms of $\delta D$ no such comparisons exist among the satellite observations." To "There are not published comparisons of $\delta D$ between available satellite observations."

Response: The text has been changed to "For $\delta D$ there are no published comparisons between these satellite observations."

page 17: line 8/9: text states "This observational discrepancy remains unresolved up to now." Are you going to resolve that discrepancy in this paper?

Response: The topic is addressed in Sect. 4.3, but not resolved. We will look into this discrepancy more detailed in a subsequent study, which is mentioned at the very end of Sect. 5.3.

**2: DATA SETS**

Can you add a statement akin to your answer to a reviewer that you consider multiple versions from the same satellite instrument because they have been published in the past and this provides context for older studies as well as future studies?

Response: The text been changed to "In this work we consider five data sets. From the SMR observations one data set is derived. Based on different retrieval versions two data sets each are obtained from the MIPAS and ACE-FTS observations. Data from the older retrieval versions have been already used in previous studies (Steinwagner et al., 2010; Randel et al., 2012) and therefore provide context for the more recent retrieval results."

2.1 Odin/SMR page 18, line 2/3 I would change" from pole to pole. This particularly concerns the boreal winter time." To "from pole to pole, mainly during boreal winter."

Response: The text has been changed to "from pole to pole, mainly during boreal winter".

2.1 Odin/SMR page 18, line 11. Change "with one orbit covering HDO, followed by one orbit covering the $H_2O$" to "with one orbit measuring HDO, followed by one orbit measuring the $H_2O$"

Response: The text has been changed to "with one orbit measuring HDO, followed by one orbit measuring the $H_2O$".

2.1 Odin/SMR page 19, line 2 change "shall" to "should" (and, can you say what the quality categories mean?)

Response: The text has been changed to "should" and definition of the quality categories has been clarified.

2.1 Odin/SMR page 19, line 17, change "drift, which are" to "drift, which is"

2.2 Envisat/MIPAS...page 20, line 21, add a comma after "(Flaud et al., 2003)" (before the "which")

2.2 Envisat/MIPAS...page 20, line 27,. Add a comma after "Steinwagner et al. (2007)" 2.2 Envisat/MIPAS...page 20, line 32, add a comma after "For $\delta D$"

Response: The points above for the data sets description have been changed in the text.

2.2 Envisat/MIPAS...page 21...line 1, this is first note of the visibility flag". Can you say what it is (as you did for the screening for SMR at the top of page 19)?

Response: A more clear description of the visibility flag has been added in the text.

2.2 Envisat/MIPAS, page 21, line 9. You can delete "However, the impact on this work is negligible."

Response: "However, the impact on this work is negligible" has been deleted in the text.

2.3 SCISAT/ACE-FTS page 22 line 21, change "utilized" to "used"

Response: The text has been changed to "used".

**3: APPROACH**

page 23 line 1 / 2 change "stratosphere. As a complement, we also use data for the upper troposphere and lower mesosphere." To "stratosphere, although we use data for the upper troposphere and lower mesosphere where available."

Response: The text has been changed to "stratosphere, although we use data for the upper troposphere and lower mesosphere where available".

page 23 line 6-10; rewrite as
(1) Calculate $\delta D$ from the individual HDO and $H_2O$ profiles, then average. Here we denote this approach as "individual."
(2) First calculate average HDO and $H_2O$ profiles, and then calculate $\delta D$. Here we refer to this approach as "separate".

Response: The text has been rewritten to:

"(1) Calculate $\delta D$ from individual HDO and $H_2O$ profiles and subsequently derive the $\delta D$ product of interest. Here we denote this approach as "individual".

(2) Derive first the product of interest separately for HDO and $H_2O$ and subsequently calculate $\delta D$. Here we refer to this approach as "separate". "

3.1 profile to profile comparisons page 25, line 14. change Haymsfield to Heymsfield

Response: The reference has been corrected.

3.1 page 25, line 10-17....I understand that you can't use equation 3 for delta-d, but can't you use it on the HDO and $H_2O$ before calculating delta-d?

Response: Yes, you can use Eq. 3 on HDO and $H_2O$ before calculating individual $\delta D$ profiles. This is also what we have done now in the revised version as stated in the text. In the original version no convolution was applied.

3.1.4, line 8, change "concerns mostly" to "impacts mainly" 3.3, page 29, line 17, delete "in particular"

Response: The text has been changed according to the suggestions.

**4: RESULTS**

4.1, page 30, This rewrite is what was needed to add more science to the comparisons. Similarly the addition on page 33/34 was needed.

Page 35, line 9, delete "typically"

Response: "typically" has been deleted.

Page 36, delta-d section, first paragraph...I'm confused by the discussion here. Some referencing to the figures for the seasonal as well as the figure 2 in the profile comparisons might help.

Response: References to the figures has been added to make the discussion clearer.

(also, in captions where you show biases, even though you say in the text, also say in the caption that the bias is A-B (or B-A) rather A vs B.

Response: The bias we present are always calculated as A-B. In the caption we write A vs. B, but there is no special meaning behind it and we consider this equivalent. Writing "minus" in the caption simply does not feel so well.

Page 38, line 21...instead of "more suited" just say "shown" (you need to say "more suited to something", not just "more suited"

Response: The text has been changed to "shown".

**5 DISCUSSION AND CONCLUSION**

page 41, line 8/9 states " This concerns primarily the comparisons to the SMR data set." which is awkward. I suggest "These quantitative differences are largest with comparisons to the SMR data."

Response: The text has been changed to "These quantitative differences are largest with comparisons to the SMR data".

page 41, line 11-14 says " In the profile-to-profile comparisons (Fig. 2), the SMR data set shows in the lower stratosphere a significantly higher depletion in $\delta D$ than the MIPAS and ACE-FTS data sets, which however is close to its lower limit, thus correspondingly the uncertainties are larger." Also, pretty awkward sentence construction. I suggest "For the profile-to-profile comparisons (Fig. 2), in the lower stratosphere the SMR data set shows significantly higher $\delta D$ depletions than for the MIPAS and ACE-FTS data sets. However, we note that this is close to the SMR lower limit, which has larger uncertainties."

Response: The text has been changed to "For the profile-to-profile comparisons (Fig. 2), in the lower stratosphere the SMR data set shows significantly higher $\delta D$ depletions than for the MIPAS and ACE-FTS data sets. However, we note that this is close to the SMR lower limit, which has larger uncertainties".

page 46, line 17/18 says " As described in Sect. 3.1.2, the differences in the vertical resolution of the MIPAS and ACE-FTS data sets can cause biases if they are not considered like in climatological comparisons." See my question above regarding section 3.1 " page 25, line 10-17". It really seems that you need to consider the resolution differences for any type of comparison. (and, if you leave this sentence alone, change "like in climatological comparison" to "as with the climatological comparisons".

Response: The text has been changed to "as with the climatological comparisons".

page 49, line 6, change "data we have at hand the least number of systematic errors" to "data currently available, fewer systematic error"

Response: The text has been changed to "data currently available, fewer systematic error".

page 49, line 10, change "principal" to "principle"

Response: The text has been changed to "principle".

[revised manuscript text omitted]

The legend entries are:

- Odin/SMR v2.1 vs. Envisat/MIPAS v5
- Odin/SMR v2.1 vs. Envisat/MIPAS v20
- Odin/SMR v2.1 vs. SCISAT/ACE−FTS v2.2
- Odin/SMR v2.1 vs. SCISAT/ACE−FTS v3.5
- Envisat/MIPAS v5 vs. Envisat/MIPAS v20
- Envisat/MIPAS v5 vs. SCISAT/ACE−FTS v2.2
- Envisat/MIPAS v5 vs. SCISAT/ACE−FTS v3.5
- Envisat/MIPAS v20 vs. SCISAT/ACE−FTS v2.2
- Envisat/MIPAS v20 vs. SCISAT/ACE−FTS v3.5
- SCISAT/ACE−FTS v2.2 vs. SCISAT/ACE−FTS v3.5

[Figure]

**Figure 10.** Tropical mean (15°S – 15°N) profiles for δD (left column), HDO (middle column) and H$_2$O (right column) for February, April, August and October given in different rows.